

# Variations in dissolved and particulate organic carbon dynamics in the lower Changjiang River on time scales from seasonal to decades

Yue Ming[1], Lei Gao[1*], Laodong Guo[2]

[1]State Key Laboratory of Estuarine and Coastal Research, East China Normal University, 3663 North Zhongshan Road,
5 Shanghai 200241, China
[2]University of Wisconsin-Milwaukee, School of Freshwater Sciences, 600 E Greenfield Ave., Milwaukee, WI 53204, USA

*Correspondence to*: Lei Gao ( lgao@sklec.ecnu.edu.cn)

**Abstract.** Time series water samples were collected monthly between November 2016 and June 2019 from the Lower Changjiang (Yangtze) River at the Xuliujing station, representing the end-member freshwater exported to the ocean, in 10 addition to monthly samples from November 2019 to January 2020 during an extreme drought event. Concentrations of dissolved (DOC) and particulate organic carbon (POC) as well as the stable isotopic composition of POC and PN (particulate nitrogen) were measured to elucidate their seasonal variations. Ultrafiltration was also conducted for selected samples to separate the bulk DOC into four different size-fractions to further examine changes in the molecular weight distribution of DOC. The seasonal variations of POC were characterized with higher proportions of autochthonous components in summer 15 and lower proportions in winter on the basis of POC contents (% of total suspended particulate matter), $\delta^{13}C$, $\delta^{15}N$, and POC/PN values. The size spectra of DOC also illustrated a distinct seasonal variation pattern, although the different responses to different seasons were also shown between HMW- (high-molecular-weight) and LMW- (low-molecular-weight) DOC compounds. The strong correlation of $\delta^{13}C$ values between HMW-DOC and POC suggested their close relationships in terms of origin and composition. Together with available literature data, DOC concentrations, POC (%), $\delta^{13}C$, and $\delta^{15}N$ all 20 showed a significant increase over the past decades, likely resulting from increase in the proportion of autochthonous organic components owing to intensified human activities and global warming in the river basin.

## 1 Introduction

Over the past several decades, the Changjiang (Yangtze) River basin has experienced complex and profound changes (Li et al., 2007; Yang et al., 2014). The global warming, at least partly owing to the intensified human activities and the extra $CO_2$ 25 outputs, has inevitably drawn its influences on various land and marine ecosystems globally (Cai et al., 2015), including the aquatic ecosystems in the Changjiang River (Wang et al., 2016), one of the largest rivers in the world. The biogeochemical cycling of carbon in this ecosystem should also be greatly modified as a feedback to the global climate change (Wang et al., 2012; 2016; Wu et al., 2007).

In addition to those above influences involving global climate changes, the anthropogenic activities should also play their 30 influences through other pathways. For example, during the recent decades, tremendous amounts of dams and reservoirs



(over 50,000) have been constructed along the main channel and tributaries over the Changjiang River basin, among which the Three Gorges Dam (TGD), accomplished and operated in June 2003, is the biggest one (Yang et al., 2011). These dams trapped tremendous sediments that should have been transported to the lower river reaches, the river mouth and the estuarine areas, or even the pelagic oceans (Deng et al., 2016; Zhao et al., 2021a). Previous studies have found that the SPM

(suspended particulate matter) concentrations in the Changjiang River water and its final fluxes to the sea have sharply decreased in response to the trapping of these dams and reservoirs (e.g., Dai et al., 2016) which, as a consequence, would largely modify the functioning, structure, and carbon cycling in aquatic ecosystems not only in the river waters but also in the adjacent estuarine and coastal oceans (Wu et al., 2007; Zhou et al., 2008; Jiao et al., 2007). Therefore, the biogeochemical cycles of carbon in aquatic environments in response to these changing hydrological regimes have long been

of great interest in the literature (Grabb et al., 2021; Xiang et al., 2021; Zhao et al., 2021b).

Due to increased industrial and agriculture developments, larger amounts of pollutants and fertilizer-derived nutrients have been discharged into the Changjiang River, resulting in the increasing nutrient concentrations (Gao et al., 2012; Li et al., 2007; Yan et al., 2003; 2010). The enhanced levers of nutrients may further stimulate the primary productivities in river waters (Li et al., 2014; Wang et al., 2021a; Zhou et al., 2008). Up to now, however, most previous studies only focused on

the changes and trends of inorganic nutrients, and how the inventories and regimes of organic matter in these aquatic environments have been altered during this important period and under the background of globe warming and the intensified human activities, is still not well addressed.

In river ecosystems, the organic matter pools are generally compositionally heterogeneous and can be roughly separated into two fractions according to their different sizes such as POC (particulate organic carbon) and DOC (dissolved organic matter),

usually by filtering the river waters using a filter with a pore size from 0.1 to 1 μm (e.g., Cai et al., 2008; Wang et al., 2012; Wu et al., 2018) . POC with large sizes and DOC with small sizes should have quite different origins, distribution and variation patterns, as well as different ultimate fates (Cai and Guo, 2009; Guo and Macdonald, 2006; Guo et al., 2009). In fact, natural organic matter in various aquatic environments should exist as a size continuum, and even within the DOC pools, different size fractions can still be separated further by techniques such as ultrafiltration, and DOC with different sizes

has been proven to still have quite different chemical properties, ages, and biogeochemical behaviors (Cai et al., 2015; Wu et al., 2018). Thus in this study, we planned to separate the organic pools, collected in the Lower Changjiang River waters at Xuliujing, into POC (> 0.7 μm) and DOC (< 0.7 μm) first, and then to separate the bulk DOC pools into the four size fractions of < 1 kDa (kilo-Dalton), 1–3 kDa, 3–10 kDa, and 10 kDa–0.7 μm, by ultrafiltrating one water samples for three more times (Gao et al., 2018; Guo and Santschi, 2007; Zhao et al., 2021b). We believe that the more detailed information

provided after ultrafiltration would further enrich and deepen our knowledge on the mechanisms and factors controlling the partitioning, dynamics, and biogeochemical behaviors of organic matters that are finally transported out by the Changjiang River.

In this study, river water samples were collected monthly at Xuliujing, and this station has been regarded as a key position well representing the Changjiang River freshwater end-member (e.g., Gao et al., 2012; Zhao et al., 2021b) . Characterizing





the organic matter in water samples collected at this station not only reflects the final transporting and cycling consequences after draining over the entire river basin (Wu et al., 2007), but also shows the concentrations, chemical properties, and fluxes of organic matter that is ready to be transported out from the Changjiang River to oceans (Gao et al., 2012; Zhao et al., 2021b).

The objectives of this study were to 1) illustrate the seasonal variations in POC and DOC (including HMW (high molecular weight) and LMW (low molecular weight)-DOC) in the Lower Changjiang River, and major factors in regulating the

dynamics of riverine POC and DOC, 2) elucidate how concentrations and chemical properties of terrestrial organic matter from the Changjiang River varied during the past decades, and 3) evaluate the influences of intensified anthropogenic activities on the seasonal and decadal variations of organic matter species from the Changjiang River.

## 2 Materials and Methods

### 2.1 Sample collection

Monthly surface water samples were collected from the main channel in the Lower Changjiang River between November 2016 and June 2019 and from November 2019 to January 2020, with a total of 35 monthly samples. All these 35 samples were with a salinity < 0.2. The sampling station at Xuliujing (31°47' N, 120°56' E, Figure 1) is the lowest freshwater end-member station, where the river begins to bifurcate into South and North Branches near the river mouth. Samples collected

between November 2019 and January 2020 represented those with extremely low river discharge during an extreme drought event in the Changjiang River basin. The lowest daily discharge, measured at the Datong station (30°46' N, 117°37' E, in the Changjiang main channel about 500 km upstream from Xuliujing, Figure 1) reached $1.05 \times 10^4$ m³/s on December 10 and 28 in 2019, compared to those lowest daily values in the other three normal dry seasons of $1.55 \times 10^4$ m³/s on February 7 in 2017, $1.21 \times 10^4$ m³/s on February 26 in 2018, and $1.16 \times 10^4$ m³/s on December 27 in 2018 (detailed discharge data are

available at http://www.cjh.com.cn/).

Right after sampling, water samples were transported to the laboratory within 3 h and filtered immediately through pre-combusted (450°C, 5 h) GF/F filters (Whatman, 0.7 μm). The suspended particles on pre-weighed filters were used to measure the concentrations of SPM, POC, particulate nitrogen (PN), and their stable isotope composition ($\delta^{13}$C-POC and $\delta^{15}$N-PN). An aliquot of the filtrates was stored at –20°C for the measurement of bulk DOC concentrations. In addition,

filtrates (400 mL each) from selected samples (August, October, and December in 2018, February, April, June, November, and December in 2019, and January 2020), including four collected from the flood season, two from the normal dry season, and three from the extreme dry season, were collected for ultrafiltration (see details below).

### 2.2 Ultrafiltration

The selected filtrate samples were further ultrafiltered to size-fractionate the bulk DOC into different size-fractions for DOM

characterization (Gao et al., 2018; Zhao et al., 2021b). Sample ultrafiltration was mostly done within 12 hours after





collection to avoid potential changes in DOM size distributions (Guo et al., 2000; Jensen et al., 2020). The bulk DOM was separated into four different size-fractions using three different disc membranes (regenerated cellulose) with different nominal molecular weight cutoffs (NMWCOs), including 1, 3, and 10 kDa (Ultracel Ultrafiltration Discs, PL series, Merck PLAC07610, PLBC07610, and PLGC07610, Amicon Bioseparations, EMD Millipore Corporation, USA). A stirred-cell
ultrafiltration device with a maximum volume of 400 mL was used (UFSC40001, Millipore Corporation, USA). During ultrafiltration, time-series permeate solutions were collected for each sample at different concentration factors (CF, defined as the ratio of the initial sample volume to the volume of retentate solution). Concentrations of DOC of the time-series permeate samples were used to quantify the abundance of permeable DOC, or low molecular weight (LMW) DOC, with sizes smaller than the NMWCO of ultrafiltration membranes (Guo and Santschi, 1996, 2007).

To quantitatively calculate the abundance of different DOC size-fractions, including the <1 kDa, 1–3 kDa, 3–10 kDa, and 10 kDa–0.7 μm, the DOC concentrations of the time-series permeate samples from each ultrafiltration were used to fit the ultrafiltration permeation model (Belzile and Guo, 2006; Guo and Santschi, 2007). The correlation coefficients ($R^2$) between log10[DOC] and log10(CF) for the 1, 3, and 10 kDa ultrafiltration were generally greater than 0.97 (or $P$-value < 0.001), supporting the constant permeation mechanism of DOM during ultrafiltration and further validating the ultrafiltration
permeation model in river samples (Zhao et al., 2021b).

To further characterize isotopic composition of colloidal or HMW-DOM of the selected river water samples, the >1 kDa retentate solutions were freeze-dried and the powder samples were collected and further processed for the measurements of $\delta^{13}C$ using IS-MS.

## 2.3 Measurements of DOC, SPM, POC and stable isotope composition

Concentrations of DOC were measured on a Total Organic Carbon Analyzer (TOC-VCPH, Shimadzu) using the high-temperature combustion method (Guo et al., 1995). Water samples were acidified with concentrated high-purity HCl to a pH < 2 before analysis. Three to five replicate measurements were made, with a coefficient of variance < 2%. Calibration curves were generated daily before the sample analysis. Concentrations of ultrapure water and certified DOC samples (from the University of Miami) were also measured every 8–10 samples to check the performance of the instrument and to ensure data
quality (Gao et al., 2020).

All filter samples were oven-dried (50–80°C) to a constant weight. The weight differences between GF/F filters and sample filters were used to calculate SPM concentrations (mg/L). Concentrations of POC and PN, and their corresponding $\delta^{13}C$ (relative to V-PDB) and $\delta^{15}N$ (relative to air $N_2$) values, were measured on the filter samples. Before POC and $\delta^{13}C$ analysis, samples were exposed to concentrated HCl vapor for at least 48 h and then dried again. Similarly, the freeze-dried HMW-
DOM powder samples were exposed to HCl vapor to remove any carbonate.

Concentrations of POC and PN and stable isotope composition $\delta^{13}C$ (relative to V-PDB) and $\delta^{15}N$ (relative to air $N_2$) of the bulk POC filters and HMW-DOM powder samples were measured on a Thermo Finnigan isotope ratio mass spectrometer (Delta plus XP, Thermo Electron Corporation, Bremen, Germany). Analytical precisions were 1% and 2% for POC and PN



measurements and were 0.1‰ and 0.2‰ for $\delta^{13}C$ and $\delta^{15}N$ respectively, as determined by replicate analysis of standards and
samples (Gao et al., 2014).

## 2.4 Statistical analysis

The seasonal Mann-Kendall test was used to analyze the decadal trends of all parameters, including the DOC concentrations,
POC (%), DOC/POC molecular ratio, $\delta^{13}C$, $\delta^{15}N$, and particulate C/N molecular ratio. Data used for trend analyses included
those from this study collected at Xuliujing and from available literature data collected at the Changjiang main channel
restricted from Datong to Xuliujing. Seasonal Mann-Kendall test for trend analysis was chosen because of its robustness to
seasonality, departures from normality, and serial dependence (Hirsch et al., 1982; Hirsch and Slack, 1984). When the
calculated $P$-value was < 0.05, it was regarded as statistically significant (increasing when the slope was positive, or
decreasing when the slope was negative) for these parameters over the last several decades. Compared to the linear
correlation analysis commonly used, the seasonal Mann-Kendall test considers the effect of seasonal variations, before
drawing the conclusion about whether the significant temporal variation trends exist (when $P < 0.05$) or not (when $P > 0.05$).
Correlation analyses were conducted using SPSS software, and $P < 0.05$ was regarded as statistically significant.

## 3 Results

### 3.1 Concentrations of SPM, POC, and DOC

The monthly discharge from the Changjiang River during the sampling period showed large seasonal variations, with the
highest monthly discharge occurred in July (e.g., $6.15 \times 10^4$ m³/s in July 2017, and $4.31 \times 10^4$ m³/s in July 2018), which are
at least twice as high as the lowest monthly discharge (e.g., $1.32 \times 10^4$ m³/s in February 2017, $1.54 \times 10^4$ m³/s in January
2018, and $1.80 \times 10^4$ m³/s in February 2019) (Figure 2). Such a large variation in the discharge should have been greatly
buffered after the completion and operation of a number of large dams and reservoirs along the Changjiang River over the
past decades (Dai et al., 2008; Deng et al., 2016; Yang et al., 2011). In addition, during the two flood seasons in 2017 and
2018, the average monthly discharge in July 2017 was about 1.4 times of that in July 2018, and the year of 2018 was a
typical "no flood in the flood season" year (Dai et al., 2008).
The SPM concentrations varied from 14 to 68 mg/L and did not show a clear seasonal variation pattern (Figure 2a). If only
the two flood seasons (June–September) were compared, the SPM concentrations in 2017 were consistently higher than
those during the same flooding months in 2018. This comparison seemly suggested that the elevated discharge values had
flushed and sustained more sediments into waters, and the flushing effect on SPM concentrations, rather than the dilution
effect might have been played by the river discharges (Creed et al., 2015). However, over the whole sampling period, SPM
concentrations did not show a simple response to river discharge on a seasonal timescale. An elevated SPM concentration
always appeared in June, the month at the early stage of flood season in the Changjiang River basin. The weak coupling



between SPM concentration and river discharge observed in the Lower Changjiang River at Xuliujing station likely resulted

from the complex landscape, diverse land cover, and multiple water sources from tributaries in the vast river basin.

POC concentrations are closely related to SPM concentrations, resulting in similar complex variation pattern between POC and SPM at Xuliujing station (Figure 2b). As shown in Figure 2, the lowest POC concentrations occurred in the extreme dry season between November 2019 and January 2020, with concentrations ranging from 57 to 61 μmol/L. The highest POC concentration was up to 157 μmol/L observed during the flood season in August 2018. In addition, a decline phase in POC

concentrations was generally found in spring (March to May) in 2017–2019.

DOC concentrations at Xuliujing varied from 119 to 209 μmol/L (155±21 μmol/L) during the sampling period (Figure 2c), which were generally higher than those POC concentrations (95±21 μmol/L) measured in the same month. However, totally different from POC, concentrations of DOC were significantly correlated with river discharge ($P < 0.001$), showing a consistent variation pattern between DOC and discharge. The elevated river discharge was always accompanied with high

DOC concentrations, suggesting that the flushing had a controlling effect on DOC concentrations in the Changjiang, similar to other world river systems, such as the Mississippi River (Cai et al., 2015) and the Yukon River (Guo et al., 2012).

DOC/POC ratios may be used to quantify the relative abundances of organic carbon between dissolved (< 0.7 μm) and particulate phase (> 0.7 μm). A simple seasonal variation pattern could not be found for DOC/POC (Figure 2d). The DOC/POC values were consistently higher than unity (1.8±0.6), and could be as high as 3.9 observed in April 2017,

indicating that DOC is the predominant OC pool in the Changjiang over the sampling period between 2017 and 2020.

**3.2. Variations in POC contents and stable isotopic composition**

Contents of POC within SPM (in %) varied from 1.9% to 4.8% (Figure 3a), showing elevated values during flood seasons and lower values during dry seasons. Although higher discharge during the flood seasons did not always lead to higher SPM or POC abundances (in mol-C/L) in river water (Figures 2a and 2b), contents of POC (in % or mg-C/g-particulate) co-varied

with discharge (Figure 3a) with significant correlation between POC% and discharge (P < 0.001).

Stable carbon and nitrogen isotopic composition ($\delta^{13}C$ and $\delta^{15}N$) of POC and PN samples also showed a similar seasonal variation pattern with discharge over the sampling period, with higher values in the flood seasons and lower values in the dry seasons (Figure 3b and 3c). Higher river discharge and SPM resulted in not only higher POC contents (POC/SPM) but also higher $\delta^{13}C$ and $\delta^{15}N$ values, indicating changes in sources of particulate organic matter or SPM between flood seasons and

dry seasons. As mentioned earlier, the river discharge during flood seasons was the highest in 2017 and higher than the flood season in 2018. If river discharge was the single most significant factor in governing the variations of POC (%), $\delta^{13}C$, and $\delta^{15}N$, one should observe higher values of POC contents and stable isotopic composition during the flood seasons in 2017 than those in the summer of 2018. However, these were not observed and instead higher values of POC(%), $\delta^{13}C$, and $\delta^{15}N$ occurred in summer 2018 (Figure 3a–3c).

Compared to marine-derived and autochthonous SPM in the Changjiang River Estuary and the adjacent shelf areas, the terrestrial SPM from the Changjiang River generally had more negative $\delta^{13}C$ values (Gao et al., 2014). As shown in Figure 3,



the δ13C values measured at Xuliujing ranged from –26.5‰ to –23.0‰, and the δ13C values were consistently lower during dry seasons or had more terrestrial signals, while those in the flood seasons were consistently higher or had more autochthonous contributions to SPM during flood season.

Values of δ15N varied from 2.2‰ during the dry season in 2016 to 7.2‰ during the flood season in 2018. Similar to DOC, POC (%), and δ13C, the higher δ15N values in summer also implied an increase in the contribution of autochthonous materials during flood seasons (Figure 3c). This seemly contradictory observation in isotopic composition between dry seasons and flood seasons suggest that changes in POM sources were not simply regulated by river discharge in anthropogenic-impacted river systems with large dams and vast population.

Contrary to the above three parameters described in Figure 3a–3c, the POC/PN ratios were lower in the flood seasons but higher in the dry seasons (Figure 3d). Low POC/PN ratios observed during summer indicated increased PN contents relative to POC resulting in decreased POC/PN ratios during flood seasons. In general, allochthonous organic materials are believed to have higher POC/PN ratios compared with autochthonous components (Gao et al., 2014; Wu et al., 2007), and the latter were characterized as having POC/PN values resemble to the Redfield ratio of 6.6. Our measured POC/PN values at

Xuliujing were generally higher than the Redfield ratio, with the highest (11.6 mol/mol) measured in February 2019 and lowest (6.3 mol/mol) appeared in July 2018. These POC/PN ratios again verified that the SPM at Xuliujing was more contributed from autochthonous sources during summer flood seasons.

### 3.3. Molecular weight distributions of DOC and δ13C values in HMW-DOC

In addition to bulk DOC concentrations, the relative abundances of the four DOC size-fractions, including the < 1 kDa, 1–3

kDa, 3–10 kDa, and 10 kDa–0.7 µm were used to evaluate changes in molecular weight distributions of DOC in the Lower Changjiang River. Furthermore, δ13C values of HMW-DOC (or the > 1 kDa size-fraction isolated by ultrafiltration) were compared with those of POC samples collected concurrently to examine the potential linkage between the two major carbon pools (POC and HMW-DOC) in the lower Changjiang River.

If the monthly discharge of $2.0 \times 10^4$ m³/s is regarded as the threshold discharge dividing all sampling months into flood and

dry seasons, the percentages of the > 1 kDa DOC (or HMZ fraction, or colloidal fraction, including the three size fractions of 1–3 kDa, 3–10 kDa, and 10 kDa–0.7 µm) were all lower than 32% in the samples from flood seasons but all higher than 43% in the samples from dry seasons (Figure 4a). In other words, the abundance of the > 1 kDa DOC in dry seasons was higher than those in flood seasons. During the extreme dry season between November 2019 and January 2020 (average monthly discharge of $1.4 \times 10^4$ m³/s), there did not exist a clear difference in the percentages of the > 1 kDa colloidal DOC compared

with other normal dry seasons (average monthly discharge of $1.8 \times 10^4$ m³/s). However, the abundances of DOC partitioned between the 1–3 kDa and 3–10 kDa size-fractions during the extreme dry season were distinctly different from those in other normal dry seasons (Table 1). For example, there was much higher DOC percentage in the 3–10 kDa fraction observed during the extreme dry season, but a higher value in the 1–3 kDa fraction in the normal dry season. These results suggested that the molecular weight distributions of DOC or DOM partitioning among different size-fractions were highly related to



changes in sources and compositions of DOM between the flood and dry seasons and between the extreme and normal dry seasons.

Comparing the average percentages in DOC among the four different size-fractions between flood and dry seasons, it is evident that the bulk DOC sampled during flood seasons contained a significantly higher abundance in the <1 kDa LMW-DOC (Figure 4b). During the dry seasons, however, the abundance of the 1–3 kDa DOC fraction (22.8 ± 8.9%) was much

higher than that during flood seasons (9.1 ± 4.1%). Although the total HMW-DOC abundance (including the 1–3 kDa, 3–10 kDa, and 10 kDa–0.7 μm size fractions) was higher in the dry season (Figure 4b), the average abundance of the 10 kDa–0.7 μm or the >10 kDa size fraction was actually higher during flood season (18.1 ± 8.2%) compared to the dry seasons (14.2 ± 4.3%). As discussed earlier, high discharges during flood seasons were generally accompanied with higher DOC concentrations (Figure 2c), DOC increased during flood seasons were mainly contributed by both the < 1 kDa LMW-DOC

and the 10 kDa–0.7 μm larger sized DOC (Figure 4b).

Overall, the molecular size spectra of DOC observed in the Lower Changjiang River at Xuliujing are highly variable with a dynamic variation among different size-fractions and between different sampling seasons (Figure 4a). During the flood seasons with generally higher DOC concentrations, DOC size spectra were characterized as the twin-peak curves. During the dry seasons with generally lower DOC concentrations, however, the DOC size spectra were shifted to the one-peak curve as

focusing into the center.

**3.4. Relationships between POC and HMW-DOC**

In general, DOC concentrations showed a close and positive response to river discharge (Figure 2c), while POC concentrations did not have a close relationship with discharges. However, the lowest POC concentrations were observed during the extreme drought event between November 2019 and January 2020 (Figure 2b), while DOC concentrations did not

show any discernable response to this extreme weather event compared with those in the normal dry seasons (Figure 2c).

$\delta^{13}$C values of the >1 kDa HMW-DOC samples (Figure 5) are in general similar to those of bulk POC samples. In addition, the $\delta^{13}$C values of the HMW-DOC were slightly higher in the dry seasons but lower in the dry seasons. Furthermore, the three $\delta^{13}$C values measured during the extreme dry season from November 2019 to January 2020 were all lower than those measured in the normal dry season (Figure 5). Regardless of flood or dry seasons, the $\delta^{13}$C values measured in the HMW-

DOC were slightly lower than those in the bulk POC, suggesting that the HMW-DOC seemed to have more terrestrial signals than POC, or, the POC pool contained more autochthonous components than the HMW-DOC (Gao et al., 2014; Wang et al., 2021b). There is a significant correlation in $\delta^{13}$C values between the HMW-DOC and bulk POC pools (Figure 5b), suggesting that the two organic carbon pools had similar sources, or there existed some rapid exchanges between the two OC pools.



## 4. Discussion

### 4.1. Potential factors influencing organic matter quantity and quality

A series of factors can influence the abundances and chemical properties of both DOC and POC pools in river waters, and thus their seasonal and even decadal variation patterns. For example, flushing or dilution induced by enhanced river discharges has been shown to affect concentrations and fluxes of different chemical molecules in river waters (Creed et al., 2015). Zhao et al. (2021b) also showed elevated Changjiang River discharges in summer probably had a flushing effect on DOC, with increasing DOC concentrations during flood seasons. On the other hand, increased river discharge always had a dilution effect on nutrients, such as $NO_3^-$ and $PO_4^{3-}$ (Gao et al., 2012). Based on our results here from more than two years of monthly sampling, increased river discharge was always accompanied with higher DOC concentrations (Figure 2c), probably implying the occurrence of flushing effect. However, POC concentrations did not show a clear relationship with river discharge (Figure 2b), suggesting that other factors also play a role in control POC variations. In addition to effects on bulk properties, sources and composition of organic carbon seemed to be also related to flushing and dilution effect. For example, during the extreme dry season between November 2019 and January 2020, POC concentrations (Figure 2b) and $\delta^{13}C$ in POC and HMW-DOC (Figures 3b and 5b) consistently showed their lowest values.

In addition to the hydrological factor, the relative contributions of allochthonous to autochthonous substances may have an influence on the abundances and chemical properties of both POC and DOC pools. For example, in spring and summer, the enhanced primary productivity could increase the contribution of biogenic organic matter contributing to the POC and DOC pools, and result in higher proportions of autochthonous organic components (Zhao et al., 2022).

Furthermore, anthropogenic activities can also show their effects. Du et al. (2021) suggested that the human activities could significantly change dissolved organic matter (DOM) pools in aquatic environments through both direct (i.e., exporting DOM from anthropogenic sources such as sewage effluent) and indirect pathways (i.e., enhancing the autochthonous production of DOM via extra nutrient loadings). Grabb et al. (2021) found that in Changjiang River waters, the $\delta^{15}N$ measured in the sewage effluent was much higher than those from the nearby soil organic nitrogen.

Human activities can also adjust the distribution pattern of water discharges and change the hydrological regime in the Changjiang River on seasonal and decadal timescales, by constructing dams and reservoirs along the main channel and the tributaries in the entire river basin. Dai et al. (2008) found that these human constructions made the Changjiang River being characterized as "no flood in flood season and no drought in dry season". If this influence, if large enough, could significantly modify the magnitudes and the occurring frequencies of the flood and drought events (Das and Meher, 2019; Walsh et al., 2020).

The constructions of dams and reservoirs not only changed the hydrological regimes in the Changjiang River, but also changed the inherent water chemistry (Ye et al., 2007; Xu et al., 2009) and phytoplankton dynamics (Zeng et al., 2006). For example, during dry seasons, water storage in the reservoirs will prolong water residence time (Sun et al., 2021), making the river ecosystem more like a lake ecosystem. During flood seasons, however, the Changjiang River was more like a real river





ecosystem (Zhang et al., 2010; Deng et al., 2016). In addition, POC and DOC dynamics might also be changed by these constructions. This is because on one hand, the prolonged residence times of river water would allow phytoplankton to bloom and take up nutrients, accumulating and increasing organic carbon biomass (Sun et al., 2021) and contributing more autochthonous and metabolism related POC and DOC (as biological leachate or as detritus). On the other hand, the prolonged residence times will make more suspended particles to settle onto the river floor, which then enhanced the water column transparence, increased the euphotic depth, and finally stimulated primary productivities (Zhang et al., 2010). All the above processes should greatly elevate the contribution of autochthonous materials to POC and DOC pools, and ultimately influence their variations on seasonal and annual or decadal timescales.

A series of studies have found that the global warming occurring over the recent decades was closely connected with the intensified human activities (e.g., Chen et al., 2019). The global warming and the increased $CO_2$ in the atmosphere may directly modify primary productivity, biological biomass, and structure, functioning and status of aquatic ecosystems, and alter the inventories and biogeochemical behaviors of carbon in large river systems (Cai et al., 2008; 2015; Guo et al., 2012; Wu et al., 2018).

### 4.2. Seasonal variations in DOC and POC

Quite different from the POC concentrations, the bulk DOC concentrations showed a significantly positive correlation with river discharge (Figure 6). It is reasonable to speculate that the flushing effect induced by increased river discharge should be the main factor that had led to the seasonal variation in bulk DOC concentrations. The difference in flushing effects between DOC and POC pools may be ascribed to their different origins or source terms. POC might be more contributed from the surface plant litter or materials overlying surface soil/sediments, while DOC would be more contributed by leachates from deep soil/sediments, autochthonous origins from *in situ* phytoplankton productions, or allochthonous origins from anthropogenic activities and effluent overflow/discharge (Wu et al., 2007; Creed et al., 2015). The scenario below can be used as an example to explain the observed deviation between POC and DOC concentrations: the floods in summer would probably have flushed more soils from deep soil horizons that contain lower POC contents (due to degradation and the presence of mineral layer) but with high DOC concentrations from deep flow paths.

In addition to bulk DOC pool, results from DOC size-fractionation by ultrafiltration provided us an opportunity to examine the behaviors of larger and smaller DOC fractions. It can be found in Figure 6 that only the < 1 kDa-DOC showed a significantly positive correlation with monthly river discharge. Unlike the bulk DOC and the LWM-DOC, the concentrations of the >1 kDa HMW-DOC was not significantly related to river discharge.

The observed seasonal variations of bulk DOC with river discharge (Figure 6) was primarily ascribed to the dynamics of LMW-DOC that accounted to more than 50% or even about 70% of the total bulk DOC (Figure 4a). The seasonal variations in the concentrations of HMW-DOC (1 kDa–0.7 μm) and POC (> 0.7 μm) pools were primarily controlled by factors, such as the enhanced primary productivities and thus increased ratios of autochthonous to allochthonous components during summer (e.g., Liu et al., 2020; Zhao et al., 2022).



As shown in Figure 5b, the $\delta^{13}C$ values were significantly correlated between POC and the HMW-DOC, suggesting similar source terms between HMW-DOC and POC. This conclusion was further supported the quasi positive correlations between concentrations of the >10 kDa HMW-DOC and POC concentrations or $\delta^{13}C$ values of POC (Figure 7).

Different from the SPM and POC concentrations, other chemical properties, such as POC (%), $\delta^{13}C$, $\delta^{15}N$, and POC/PN, all showed distinct seasonal variations (Figure 3). These parameters have been used to differentiate organic and inorganic matter sources, including those from terrigenous detritus, soils, plants, phytoplankton biomass, and anthropogenic inputs (Careddu et al., 2015; Liu et al., 2009; 2019b; Tesi et al., 2007; Zhang et al., 2020). Previous studies showed that in the Changjiang River Estuary, the $\delta^{13}C$ values in phytoplankton biomass in the adjacent marine waters typically ranged from −21‰ to −19‰, while those for POC from the Changjiang River were far more negative, usually between −27‰ and −25‰ (Wang et al., 2021b; Wu et al., 2007; Zhang et al., 2007).

Autochthonous DOM in river waters mainly comes from *in-situ* biological productions, whereas allochthonous DOM is largely from the decomposition of dead organisms, such as plants, and domestic, industrial and agricultural sewages (e.g., Liu et al., 2020). In the Lower Mississippi River, Cai et al. (2015) reported that the $\delta^{13}C$ values in the DIC (dissolved inorganic carbon) pool was much higher (−9‰ and −6‰) than those measured in bulk DOC and HMW-DOC (−27‰ and −24‰) and in POC (−28‰ and −25‰). Wang et al. (2016) also found that the $\delta^{13}C$ values of DIC measured in March and August were all falling in the range between −8.7‰ and −7.6‰ in the Changjiang River, which are much higher than those in the bulk POC collected from the same area (Gao et al., 2012). Sato et al. (2006) reported that the $\delta^{13}C$ values in the autochthonous POC of Tokyo Bay were apparently correlated with the $\delta^{13}C$ values in DIC, and the latter had much higher $\delta^{13}C$ values (−10‰ and 0‰) than those of allochthonous and terrestrial POC with $\delta^{13}C$ values close to −26.5‰. Based on the n-alkane data, Zhao et al. (2022) found that the autochthonous organic carbon contributed more to the total organic matter during flood seasons (55%) than dry seasons (46%) in the Changjiang River. Qu et al. (2019) found that in the Huanghe (Yellow) River Estuary, the $\delta^{13}C$ and $\delta^{15}N$ values in autochthonous POC from phytoplankton biomass were also generally less negative than those measured in terrestrial organic materials. All the above results strongly supported the viewpoint that the rhythmically higher POC (%), $\delta^{13}C$, and $\delta^{15}N$ values and low POC/PN ratios during flood seasons observed in the lower Changjiang River (Xuliujing) could be primarily ascribed to the higher proportions of autochthonous materials in bulk POC in summer.

Overall, the seasonal variations of parameters describing the chemical properties of POC should not have been mainly caused by the flushing or dilution effects. If river discharge had really played a dominant role in governing the POC chemical properties, the POC concentrations should also have displayed a clear seasonal variation. However, this is not occurring (Figure 2b). In addition, it is quite possible that the distinct seasonal variations in DOC concentrations (Figure 2c) should also, at least partly, be ascribed to the seasonal variations of the autochthonous/allochthonous ratios, because, as pointed out before, the larger DOC size-fractions (e.g., >1 kDa or > 10 kDa) were more closely related to POC, rather than to bulk DOC or LMW-DOC (Figures 5–7).



### 4.3. Decadal trends

Over the past 60 years, a significant decadal trend of annual river discharge from the Changjiang River was not clearly found (Figure 8a), while a sharply decreased trend of the annual SPM load was observed (Figure 8b). The decreased SPM concentrations and fluxes had been proven to be mainly ascribed to the trapping by dams and reservoirs along the river during the recent decades (e.g., Yang et al., 2011). Since 2003 when the data began to be available, we found that the grain size of SPM measured at Datong became coarser and coarser, from about 0.01 mm in 2003 to 0.02 mm in 2020 (Figure 8b),

suggesting that finer SPM was preferentially trapped in reservoirs and dams. Changes in SPM size observed here might also have an influence on the concentrations and chemical properties of POC or even those of the larger-sized DOC in the lower Changjiang River.

Consistent with the sharply decreased loads of SPM transported by the Changjiang River, Gao et al. (2012) also found that the concentrations and export fluxes of POC both largely decreased during recent decades, especially after the operation of

TGD in June 2003. In this study, we focused more on variations in the concentrations and fluxes of POC and DOC after June 2003 (Figures 9 and 10). Using our new data reported here combined with available literature data, although the seasonal Mann-Kendall test showed a significantly decreasing trend of POC concentrations from 2003 to 2019, the decreasing slope value was very small (0.02 μmol/(L·yr), see Figure 9a). During the same period, the seasonal Mann-Kendall test verified that the concentrations of DOC had a significantly increasing trend with slope values much higher by more than one order of

magnitude (0.21 μmol/(L·yr), shown in Figure 10a). The above results again suggested that the origins and subsequent consuming and replenishing processes of POC and DOC over the Changjiang River basin were actually two different stories, not only on a seasonal timescale (Figure 2b vs. Figure 2c), but also on a decadal timescale. Based on the limited data, the annual fluxes of DOC exported by the Changjiang River seemed to show the increasing trends (Figure 10b and 10c), consistent with its concentration trend (Figure 10a).

The seasonal Mann-Kendall test further demonstrated that the content of POC (%) had increased from about 1% before the construction of TGD in 1994 or 2003 to higher than 2% after 2003 (Figure 11a). Previous studies found that POC (%) in surface sediments was generally about 1% (e.g., Zhou et al., 2006). This reference value is very important and it tells us that SPM regimes in the Changjiang River probably have completely changed over the past decades. The constructed dams and reservoirs may preferentially trap allochthonous/terrestrial SPM containing a lower POC, resulting in higher POC (%) in the

remaining SPM and increased contribution from autochthonous materials. In addition, the global warming and intensified human activities may also elevate proportions of the autochthonous and biologically-derived substances in river waters.

The seasonal Mann-Kendall test also verified the significant increasing trends of $\delta^{13}C$ and $\delta^{15}N$ during the same period (Figure 11c and 11d) in addition to POC (%). However, a significant trend was not found for both DOC/POC and POC/PN ratios (Figure 11b and 11e), probably resulting from the insufficient database. From their seasonal variations, the higher POC

(%), $\delta^{13}C$ and $\delta^{15}N$ (Figure 3a–3c), and DOC concentrations (Figure 2c) all indicated higher proportions of autochthonous organic substances in river waters. If this conclusion can be extended to explain their decadal variations, the organic matter





regimes in Changjiang River waters may have changed from a "primitive" one (more contributed by allochthonous materials) to a "civilized" one (more contributed by autochthonous materials), under the rapid economic and social developments in China. On the basis of our dataset here, DOC concentrations, POC (%), $\delta^{13}$C, and $\delta^{15}$N could serve as reliable and sensitive
proxies to trace the changes in chemical and biological regimes in the Changjiang River ecosystems on both seasonal and decadal timescales.

**5. Conclusions**

Longer term monthly samples were collected from the Lower Changjiang River at the Xuliujing station for the measurements of concentrations of POC and DOC in river waters and POC (%), $\delta^{13}$C, $\delta^{15}$N, and POC/PN ratios in suspended
particle matter. In addition, ultrafiltration was conducted in selected monthly samples to examine DOC size partitioning over the sampling period.

Our results show that SPM and POC concentrations, as well as DOC/POC ratios did not show a straightforward seasonal variation pattern. However, DOC concentrations were significantly correlated with river discharge, with higher values during flood seasons and lower values during dry seasons. Similar to DOC, concentrations of SPM and POC and other intensive
properties, such as POC (%), $\delta^{13}$C-POC, $\delta^{15}$N-PN, and POC/PN, also showed simple seasonal variations, with higher values of POC (%), $\delta^{13}$C, and $\delta^{15}$N, and lower POC/PN ratios in flood seasons. During dry seasons, lower values of POC (%), $\delta^{13}$C and $\delta^{15}$N, and higher POC/PN ratios were observed.

Results of DOC size-fractionation using ultrafiltration show that, during the flood seasons, the bulk DOC pool was more contributed by the <1 kDa LMW-DOC and the 10 kDa–0.7 µm HMW-DOC, while during the dry seasons, bulk DOC was
more contributed by the intermediate sized DOC (i.e., the 1–3 kDa and 3–10 kDa). Although DOC concentrations were generally significantly correlated with river discharge, the significant relationship was mainly attributed to the LMW-DOC (< 1 kDa). Similar isotopic composition ($\delta^{13}$C values) between POC and the >1 kDa HMW-DOC suggested that the origins and behaviors of the HMW-DOC were resembled to those of bulk POC. Higher primary productivities during summer seasons probably led to higher proportions of autochthonous organic matter, and thus the elevated POC (%), $\delta^{13}$C (in both
POC and HMW-DOC), and $\delta^{15}$N values as well as the lower POC/PN values.

Based on the seasonal Mann-Kendall test using our data combining with available literature data, we found that the POC (%), $\delta^{13}$C, and $\delta^{15}$N values, as well as bulk DOC concentrations all showed a significantly increasing trend during the past decades, especially after the completion of the TGD in 2003. In addition, global warming might have elevated river water temperature, and the increasingly intensified human activities, such as construction of dams and changes in land-cover,
might have reduced river SPM loadings that in turn allowed more sunlight penetrating into river water, promoting photosynthesis in the water column. Furthermore, human activities had directly discharged large amounts of nutrients into river systems, favoring algal blooms. Thus, decreased SPM loadings and increased primary productivities likely led to a higher autochthonous contribution relative to allochthonous organic matter in the lower Changjiang River. The significant



decadal changes are clear in the parameters reported here and likely reflected the ecosystem evolution over the whole river
basin although further studies are needed to tackle sources and their apportionment of river waters and different organic
matter pools.

**Acknowledgments**

We thank Lingbin Zhao for her help during the field work. This study was funded by Science and Technology Commission
of Shanghai Municipality (19230711900), the National Natural Science Foundation of China (U2040216), and the
Fundamental Research Funds for the Central Universities of China. The data in this study will be made available upon
request to the corresponding author. The authors declare that there is no conflict of interests regarding the publication of this
article.

**Author Contributions**

Conceptualization, Methodology, Formal Analysis, Investigation, Data Curation, Writing – Original Draft Preparation, and
Writing – Review & Editing: Yue Ming and Lei Gao; Visualization: all authors; Project Administration: Lei Gao; Funding
Acquisition: Lei Gao.

**Availability of data and material**

Data used in this study can be found and accessed online.
(at https://doi.org/10.6084/m9.figshare.19207581.v1)

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




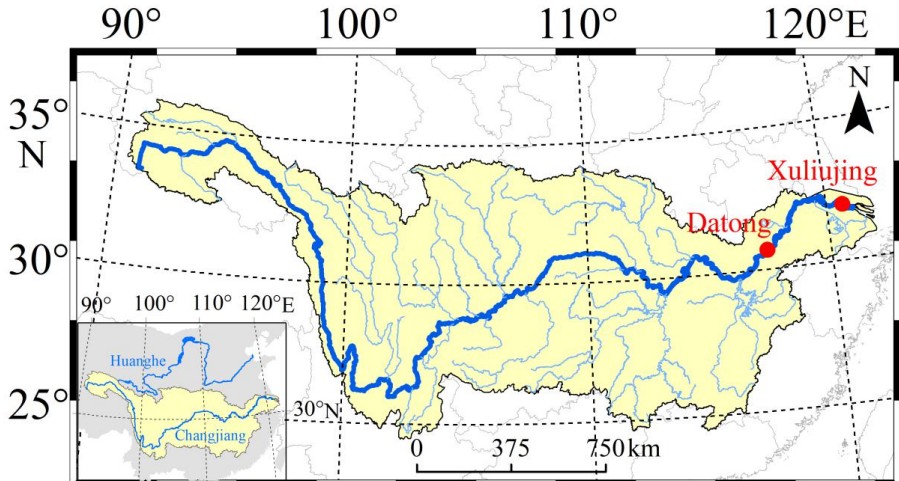

**Figure 1: Map of the Changjiang River basin and its tributaries (the blue lines) as well as locations of Datong (where data of river discharge were obtained) and Xuliujing (where the water samples were collected) stations in the lower reach.**

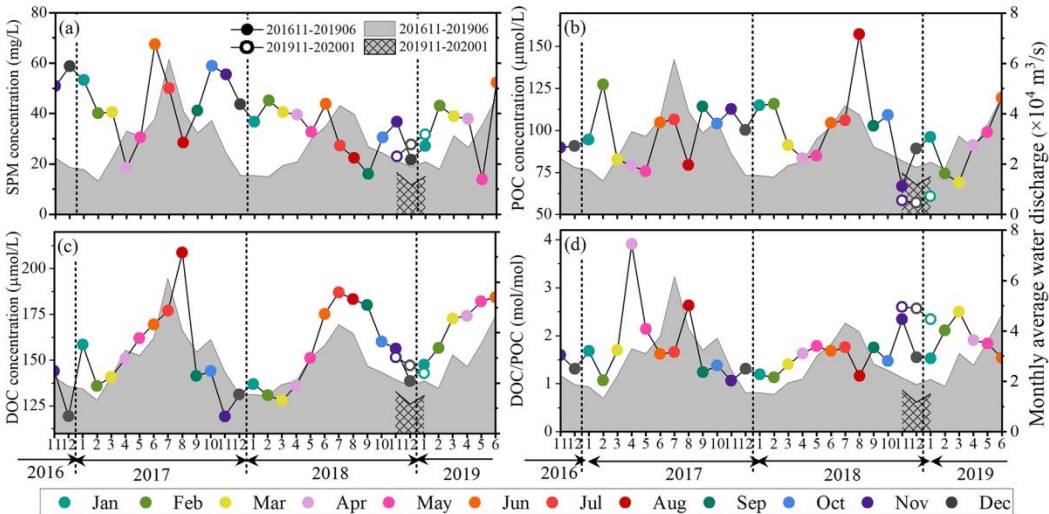


**Figure 2: Variations in concentrations of SPM, POC, and DOC and DOC/POC ratio at Xuliujing station in the Lower Changjiang River, compared to monthly discharges at Datong (shown as the shaded area based on data from http://www.cjh.com.cn/) during the periods between November 2016 and January 2020.**




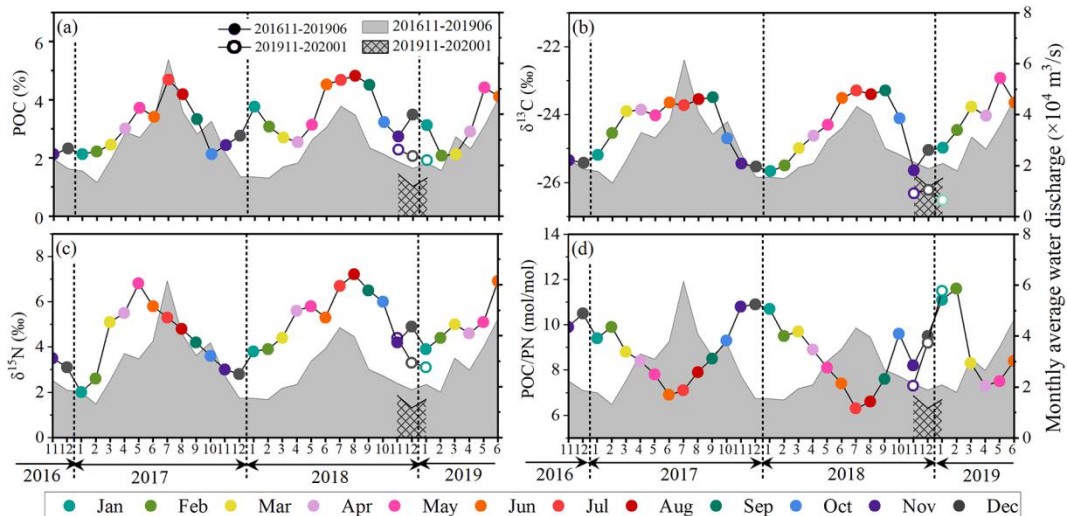

**Figure 3: Variations of POC (%), δ¹³C-POC, δ¹⁵N-PN, and POC/PN ratios in the Lower Changjiang River at Xuliujing between November 2016 and January 2020, with discharge shown by grey shade (data from http://www.cjh.com.cn/).**

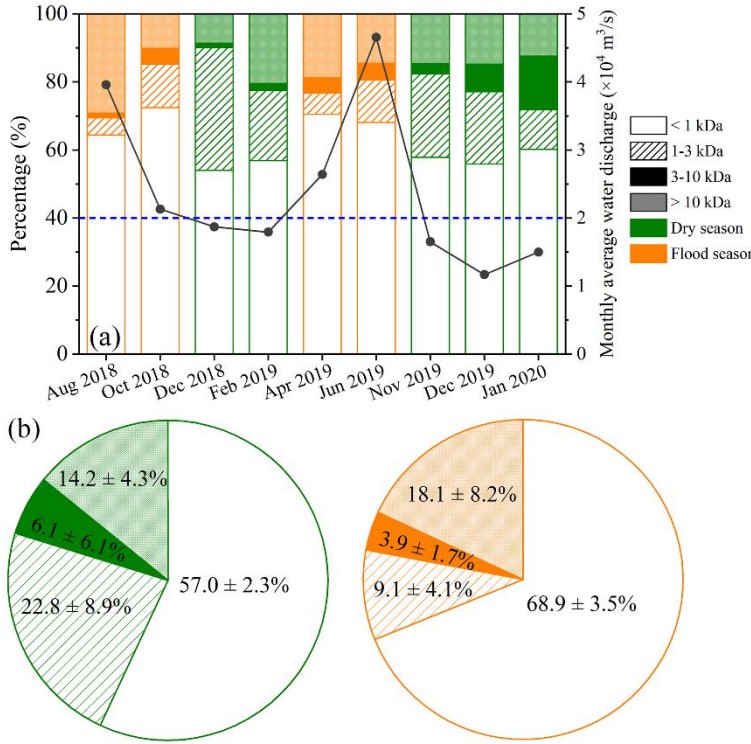

**Figure 4: (a) Variations in the size distributions of bulk DOC among the <1 kDa, 1−3 kDa, 3−10 kDa, and >10 kD fractions with the blue dashed line highlighting the monthly average discharges of 2.0 × 10⁴ m³/s between the flood seasons (above the dashed line) and dry seasons with discharge below the dashed line). (b) Comparisons in the average molecular weight distribution of DOC between the dry (left) and flood (right) seasons.**



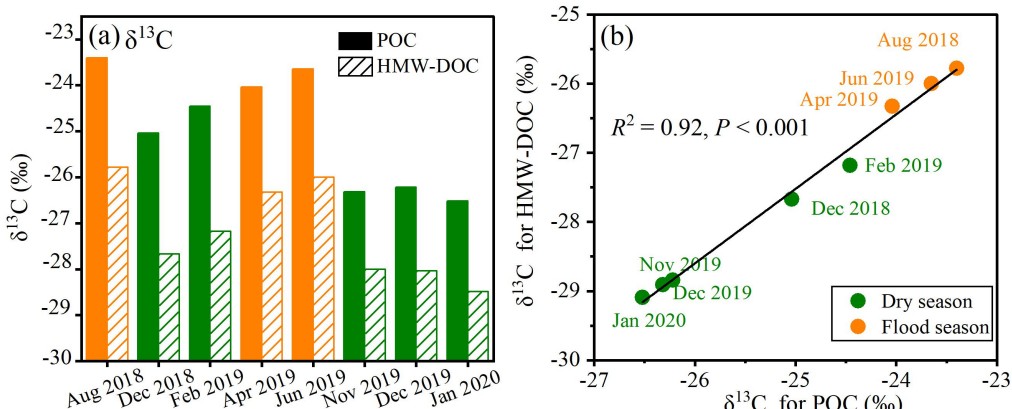

**Figure 5: (a) Comparisons in the δ¹³C values between the HMW-DOC and bulk POC pools. (b) The correlation of δ¹³C values**
**between the HMW-DOC and POC pools in the Lower Changjiang River.**

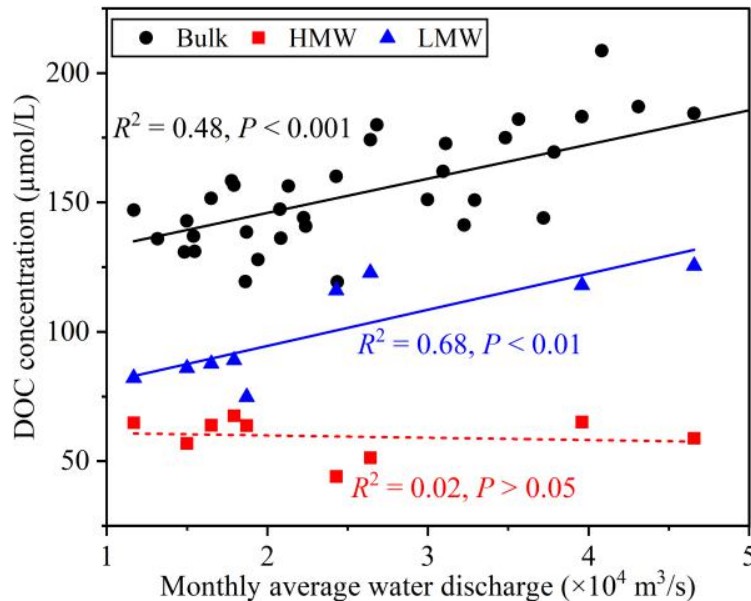

**Figure 6: Correlations between monthly discharge and the concentrations of bulk DOC, HMW-DOC (> 1 kDa), and LMW-DOC**
**(< 1 kDa) in the Lower Changjiang River.**




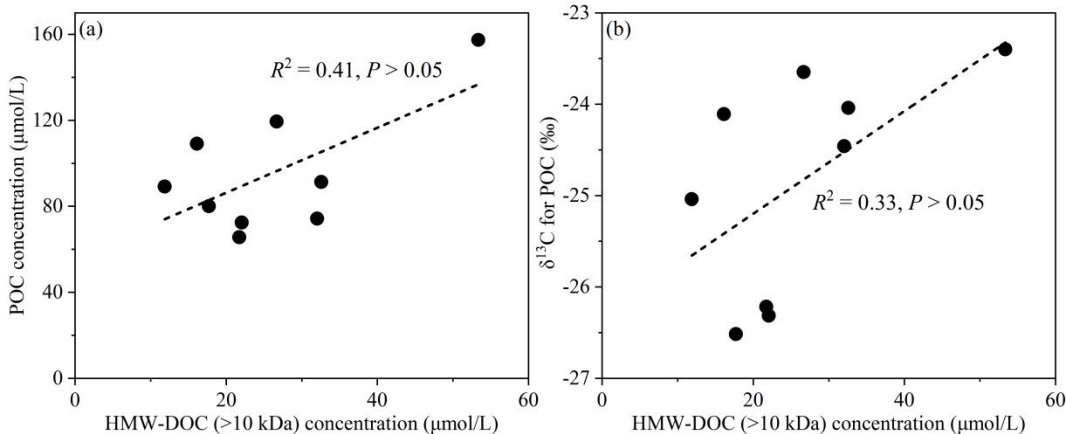

**Figure 7: (a) The correlations between the concentrations of POC and the > 10 kDa HMW-DOC, and (b) between the δ¹³C values of POC and the concentrations of the > 10 kDa HMW-DOC.**

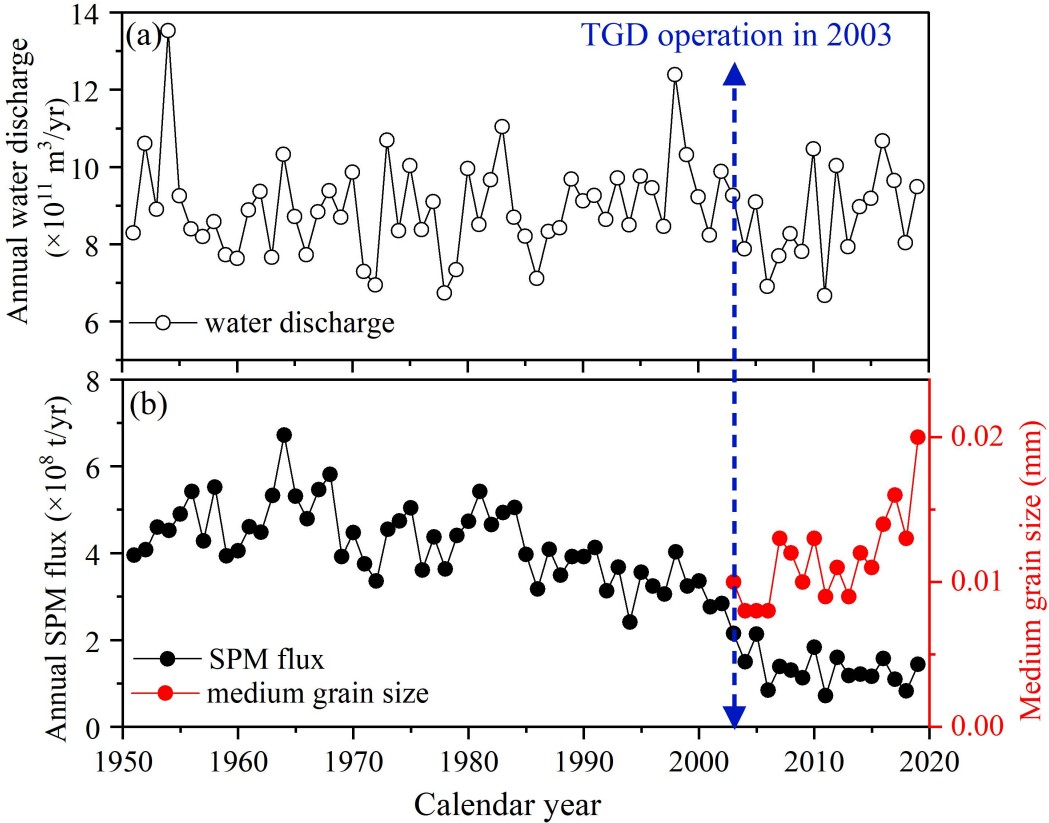


**Figure 8: Variations in annual discharge, SPM fluxes, and the SPM grain size measured at Datong in the lower Changjiang River between 1950 and 2020 (data from http://www.cjh.com.cn/). The blue dashed line represents the time when the three Gorges Dam (TGD) became operational.**





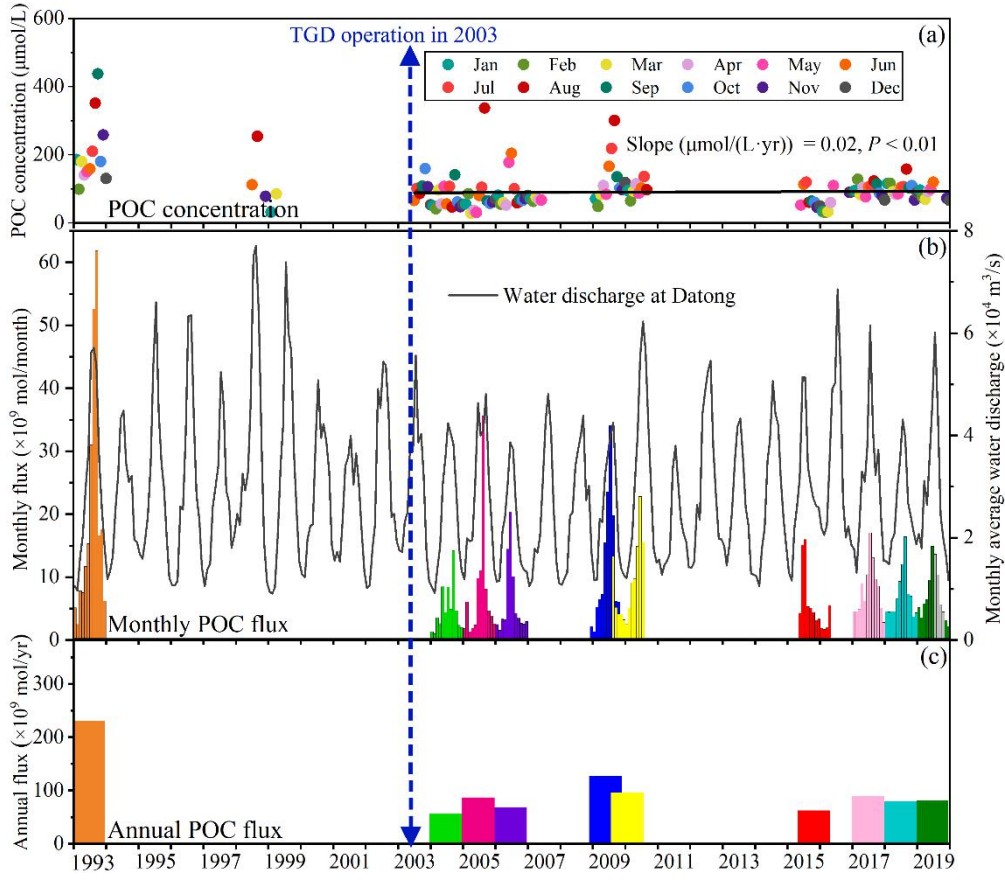

**Figure 9: Variations in concentrations (a) and monthly (b) and annual fluxes of POC in the lower Changjiang River restricted from Datong to Xuliujing between 1993 and 2019. The linear line in panel (a) indicates the significant decreasing trend of POC concentrations from 2003 to 2019 after conducting the seasonal Mann-Kendall test. The POC concentration data are from Cai and Han (1998) at Datong (January 1993 to December 1993), Duan et al. (2008) at Datong (June 1998 to March 1999), Lin (2007) at Xuliujing (June 2003 to December 2006), Zhang et al. (2014) at Datong (January 2007 to May 2007), Bao (2013) at Xuliujing (January 2008, August-October 2009, and July-August 2010), Wang et al. (2012) at Datong (January 2009 to December 2009), Gao et al. (2012) at Xuliujing (September 2009 to August 2010), Liu et al. (2019a) at Xuliujing (May 2015 to April 2016), and this study at Xuliujing (November 2016 to December 2019).**



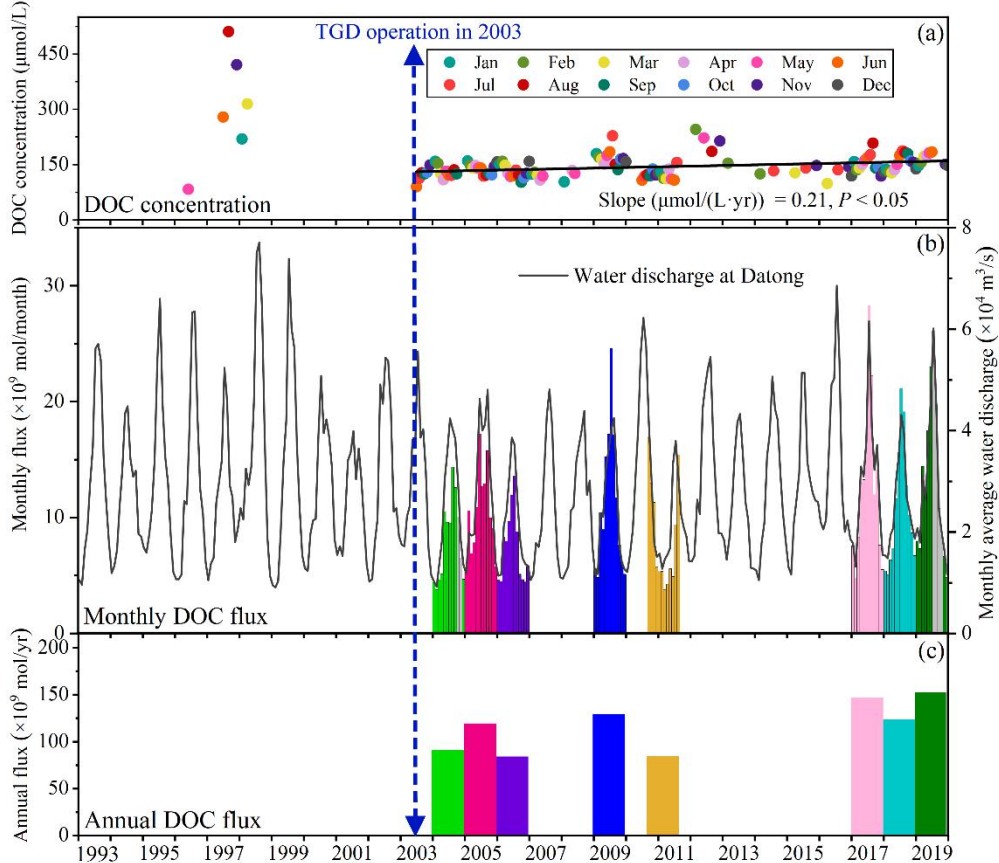

**Figure 10: Variations of DOC concentrations (a) and monthly (b) and annual fluxes (c) in the lower Changjiang River restricted from Datong to Xuliujing Between 1993 and 2019. The linear line in panel (a) indicates the significant increasing trend of DOC concentrations from 2003 to 2019 after conducting the seasonal Mann-Kendall test. The DOC data are from Wu et al. (2007) at Nantong (April-May 1997 and 2003), Duan et al. (2008) at Datong (June 1998 to March 1999), Lin (2007) at Xuliujing (June 2003 to December 2006), Zhang et al. (2014) at Datong (January 2007 to May 2007), Bao (2013) at Xuliujing (January 2008, August-October 2009, and July-August 2010), Wang et al. (2012) at Datong (January 2009 to December 2009), Wang (2018) at Xuliujing (June 2010 to July 2011), Xing et al. (2014) at Xuliujing (February 2012 to November 2012), Gao et al. (2020) at Xuliujing (February 2014 to July 2016), Zhu (2020) at Xuliujing (March 2015), Zhang et al. (2018) at Xuliujing (November 2015), and this study at Xuliujing (November 2016 to December 2019).**







**Figure 11: Variations of POC (%), DOC/POC ratios, δ¹³C, δ¹⁵N, and POC/PN ratios in the lower Changjiang River between Datong and Xuliujing between 1993 and 2019 (river discharges are provided in the background in grey). The linear lines in panels**



(a), (c), and (d) indicate the significant increasing trends of POC (%), $\delta^{13}$C, and $\delta^{15}$C in the recent decades based on the seasonal Mann-Kendall test. Data plotted here are from Duan et al. (2008) at Datong (June 1998 to March 1999 in panels a and b), Wu et al. (2002) at Nantong (July 1996 to July 1999 in panels c and d), Lin (2007) at Xuliujing (June 2003 to December 2006 in panels a, b and e), Zhang et al. (2014) at Datong (June 2006 to May 2007 in panels a and b), Mao et al. (2011) at Xuliujing (July 2007 and December 2007 in panels c), Xu et al. (2011) at Nantong (April 2008 to March 2009 in panels c and e), Wang et al. (2012) at Datong (January 2019 to December 2019 in panels a, b, c and e), Gao et al. (2012) at Xuliujing (September 2009 to August 2010 in panels a, c, d and e), and this study at Xuliujing (November 2016 to December 2019).

**Table 1. Percentages of different DOM size fractions and their standard deviation (±SD) in surface water collected at the Xuliujing Station between the flood, dry, and the extreme dry seasons.**

| Size fraction | < 1 kDa | 1-3 kDa | 3-10 kDa | 10 kDa-0.7 μm |
|---|---|---|---|---|
| Flood season | 68.9% ± 3.5% | 9.1% ± 4.1% | 3.9% ± 1.7% | 18.1% ± 8.2% |
| Normal dry season | 55.4% ± 2.0% | 28.4% ± 11.0% | 1.7% ± 0.5% | 14.5% ± 8.4% |
| Extreme dry season | 58.0 % ± 2.2% | 19.1% ± 6.7% | 9.0% ± 6.4% | 13.9% ± 1.3% |