# Peer review of "Variations in dissolved and particulate organic carbon dynamics in the lower Changjiang River on time scales from seasonal to decades"

_EGUsphere, 2022_

## Referee Comment (RC3)

**Comments on egusphere-2022-30**

Major comments:

This paper presents DOC and POC data at the Xuliujing Station of the Yangtze River over three years. These data are very valuable to explore the seasonal and long-term variations and controls of organic carbon exported by the Yangtze to coastal oceans. However, the analysis is not complete or thorough. Some of the major conclusions are not supported by the presented data.

(1) The authors conclude that higher $^{13}$C-POC in summer was due to autochthonous production in upstream intensified by human activities (e.g., the Three Gorges Dam). However, I am not convinced on this point. The authors should also consider other factors potentially affecting $^{13}$C signals of organic carbon. First, Poyang Lake and Dongting Lake are very important sources of water to the main channel of Yangtze (20-30%). These large lakes also contribute organic carbon to the main channel. Second, summer features high discharge and high sediment load, which does not favor autochthonous production (authors also stated this point, e.g., in line 287-288). In fact, a lot of studies have shown high phytoplankton activities in the Yangtze in winter or spring when flow and suspended sediment content are low. Third, Xuliujin is the last station before Yangtze enters the coastal ocean. Hence, the tidal influence is significant. Is it possible that autochthonous production in coastal ocean cause the higher $^{13}$C values in summer? How does estuarine process affect organic carbon biogeochemistry of the river?

(2) The authors conclude on a significant increase of POC, $^{13}$C-POC and $^{15}$N-PN over the past decades based on the literature and their own data, which was attributed to increase in the proportion of autochthonous organic components owing to intensified human activities and global warming in the river basin. However, these data are from three different stations (Datong, Nantong and Xuliujing), and the distance between Datong and Xuliujing stations could be as high as > 500 km. In particular, the Xuliujing station is also likely affected by autochthonous production in the estuary. It seems that the increasing trend of POC and isotopes is likely caused by geographic rather than temporal variations. Also, there is a large gap linking the observed variations with human activities and global warming.

I think the above problems are critical to the validness of the conclusions.

Specific comments

Abbreviations (not full name) should be shown in brackets in the text for the first time.

Line1-2: the words "variations" and "dynamics" in title are replicate.

Line 8-10: the abstract did not show research background or scientific questions. Research significance is not shown in the Abstract either.

Line 19-21: the increasing trend need to be reconsidered based on the same station; contribution of autochthonous component to DOC and POC should consider the influence of estuary phytoplankton dynamics and tidal activities.

Line 39-47: authors stated that biogeochemical cycles of carbon in aquatic environments had long been of great interest in the literature, but did not state the existing work on carbon variations under the influences of climate change and anthropogenic activities. And the knowledge gap deducted from existing work is not clear either in this paper.

Line 56-59: these sentences are research methods.

Section 4.1: the potential factors influencing organic matter quantity and quality were only discussed using their own data when referring to flushing and dilution effects (e.g., lines 263-265, 267-268), but all the others were repeating the conclusions that have been reported by literatures without discussing the major findings of the present study. Therefore, it is hard to tell whether human activities, global warming and autochthonous production did show their effects in the present study. I suggest that the authors concisely summarize their major findings that directly answer their main research questions and focus on explaining and evaluating what they found.

Section 4.2: I agree that the seasonal variations of POC and its isotopes may be related to the ratios of autochthonous to allochthonous components, but I am reserved on that the autochthonous signal of

POC at Xuliujing station is related to the upstream reservoir constructions (e.g., the Three Gorges Dam) which is like >1500 km far away. What about the influences of autochthonous production in upstream lakes (e.g., Dongting and Poyang Lakes)? And is it possible that the POC quantity and quality are influenced by the phytoplankton dynamics in the estuary or coastal ocean where autochthonous production is strong in summer?

Line 309-311: is there any data supporting the source of POC from deep soils? Why are waters from deep flow paths high in DOC concentration? Can belowground water influence DOC concentration?

Section 4.3: authors tried to show the decadal trend of SPM, DOC, POC and their isotope signals using reported data and literature data. This is a great idea, however, these data are from three different stations (Datong, Nantong and Xuliujing), of which the Xuliujing station is likely severely influenced by estuary phytoplankton dynamics and tidal activities. More importantly, the increasing trend of POC, $^{13}$C-POC and $^{15}$N-PN in Figure 11 is very likely caused by different stations (higher values in Xuliujing station) instead of time. I'm afraid that the decadal trends need to reconsideration.

Figure 1: add dam positions.

Figures 9 and 10: when investigating temporal variations, using data from the same station should be more compellent.

Figure 11: add legend of points including station name.

---

## Author Comment (AC1)

Response to Comments (paper # EGUsphere-2022-30)

**Variations in dissolved and particulate organic carbon in the lower Changjiang River on time scales from seasonal to decades**

Yue Ming[1], Lei Gao[1]*, and Laodong Guo[2]

1. State Key Laboratory of Estuarine and Coastal Research, East China Normal University, 3663 North Zhongshan Road, Shanghai 200241, China
2. University of Wisconsin-Milwaukee, School of Freshwater Sciences, 600 E Greenfield Ave., Milwaukee, WI 53204, USA

Correspondence to: lgao@sklec.ecnu.edu.cn

We gratefully thank Reviewer for your time and valuable comments on our manuscript. We have carefully considered these comments and revised the manuscript accordingly. Our response to reviewers' comments point by point is given below. The original comments (in blue text) are also provided followed by our detailed response (regular font size).

**Reviewer #1:**
This study investigated the monthly variations of particulate and dissolved organic carbon (POC, DOC) in the lower Changjiang River. The used stable carbon isotope approach combined with concentration measurement and ultrafiltration technique to elucidate the sources and seasonal variations of POC and DOC in the river as related to the discharge and possible influence of human activity and climate changes in the river. This is an interesting study and it provides valuable new information for our understanding of the sources and dynamics of terrestrial organic matter transported by the Changjiang River which is one of the largest rivers in the world and has great influence on the carbon cycling and biogeochemical processes in the East China Sea. Overall, it is a nice paper and I like to see its publication in Biogeosciences after some minor to moderate revision. The following lists some suggestions.

**Response:** Thank you very much for your positive comments.

1. The results indicated that the concentrations of SPM (suspended particulate matter) didn't show good correlations with river discharge and seasons during 2016-2019. Could this be related to the sampling variations? I expect that SPM is not like DOC, it may not be distributed uniformly in the river. It is not mentioned how much water was filtered for SPM. Was the water volume consistent used for all SPM sample collections? Any duplicate SPM samples were collected? This should be an easy thing to do.

**Response:** Thank you for the comments. In the Changjiang River, it was generally regarded that higher river discharge in the summer should always be accompanied with higher SPM concentrations, largely due to the enhanced soil erosion induced by elevated water discharges in summer (Dai et al., 2016). This situation was indeed true a few decades ago because of the intensive deforestation over the river basin. However, due to the increased damming effect and decreased deforestation over the river basin, the difference in SPM concentrations between flood and dry seasons in the Changjiang River became smaller and smaller (see Figures D11 and D12, the letter D means the reference of Dai et al. [2016], similarly hereafter). Therefore, based on our observed SPM data between 2016 and 2020 (see data in this manuscript) and those reported for the same sampling locations in 2009–2010 (see Figure G4a of Gao et al. [2012]), no significant difference in SPM concentrations was found between flood seasons and dry seasons in the Changjiang River during the recent years from 2009 to 2020.

[Figure]

Figure D11. Daily SSC (suspended sediment concentration) and water discharge in different flood years: A) 1964; (B) 1973; (C) 1983; (D) 1998; (E) 2010.

[Figure]

Figure D12. Daily SSC and water discharge during different drought years: (A) 1963; (B) 1978; (C) 2011.

[Figure]

Figure G4a. Variations of the monthly average values (±standard deviation) of SPM abundance measured at station #4 during the sampling period September 2009 to August 2010.

Regarding the 35 samples involved in this study, the water volumes used for filtration were based on the specific SPM concentrations, ranging from 150 mL to 400 mL, with an average value of 266 ± 126 mL. In order to assess the reproducibility and repeatability, ten duplicate filtrations, with the same water volume (250 mL), were conducted for a Xuliujing sample collected in July 2021. The obtained SPM

concentrations were $30.6 \pm 2.6$ mg/L ($n = 10$) with a relative standard deviation of 8.4%. This information has been added into the revised manuscript.

References:

Dai, Z., Fagherazzi, S., Mei, X., and Gao, J.: Decline in suspended sediment concentration delivered by the Changjiang (Yangtze) River into the East China Sea between 1956 and 2013, Geomorphology, 268, 123-132, https://doi.org/10.1016/j.geomorph.2016.06.009, 2016.

Gao, L., Li, D., and Zhang, Y.: Nutrients and particulate organic matter discharged by the Changjiang (Yangtze River): Seasonal variations and temporal trends, J. Geophys. Res.: Biogeo., 117, G04001, https://doi.org/10.1029/2012JG001952, 2012.

2. Small water volume (400 ml) was used for the ultrafiltration in this study. Did the efficiency of the ultrafiltration method using different pore sizes filters have been tested using standard compounds of knowing molecular weight? I think the authors probably did. If so, please add this information in the Method Section.

**Response:** The stirred cell ultrafiltration unit used in this study had a maximal volume of 450 mL. In addition, compared to seawater, river water normally contains higher DOC and DOM. For our measurements of DOC and DOM here, 400 mL is sufficient, which was consistently used for ultrafiltration. Regarding the pore-size or nominal molecular weight cutoffs (NMWCOs) of membranes used in this study, those from manufacture's specification or cutoff ratings were given/listed in the manuscript for easy comparisons with other studies and to avoid confusing since the actual NMWCOs could be higher than those of manufacture's rated cutoffs (e.g., Xu and Guo, 2017; Zhou and Guo, 2015). For the 1 kDa membrane, we used vitamin $B_{12}$ solution (MW = 1.3 kDa) to check membrane's integrity, as described in previous studies (Guo et al. 2000; Xu and Guo, 2017). On average, <15% of vitamin $B_{12}$ was measured in the <1 kDa ultrafiltrate. For better comparisons, the same ultrafiltration membranes and protocols (ultrafiltration permeation model or concentration difference) are highly recommended. And this technical issue has been discussed in details in Zhao et al. (2021). Following reviewer's suggestion, related sentences describing this technical issue have been added into the Method section of the revised manuscript.

References:

Guo, L., Wen, L.-S., Tang, D., and Santschi, P. H.: Re-examination of cross-flow ultrafiltration for sampling aquatic colloids: evidence from molecular probes, Mar. Chem., 69, 75-90, https://doi.org/10.1016/S0304-4203(99)00097-3, 2000.

Xu, H. and Guo, L.: Molecular size-dependent abundance and composition of dissolved organic matter in river, lake and sea waters, Water Res., 117, 115-126, https://doi.org/10.1016/j.watres.2017.04.006, 2017.

Zhao, L., Gao, L., and Guo, L.: Seasonal variations in molecular size of chromophoric dissolved organic matter from the lower Changjiang (Yangtze) River, J. Geophys. Res.: Biogeo., 126, e2020JG006160, https://doi.org/10.1029/2020JG006160, 2021.

Zhou, Z. and Guo, L.: A critical evaluation of an asymmetrical flow field-flow fractionation system for colloidal size characterization of natural organic matter, J. Chromatogr. A, 1399, 53-64, https://doi.org/10.1016/j.chroma.2015.04.035, 2015.

3. On line 114, please state what is IS-MS for the first time.

**Response:** Sorry for the typo. It should read isotope-ratio mass spectrometer (IR-MS).

4.The Results Section can be more focused on the results only, some discussion sentences can be moved to Discussion Section.

**Response:** We agree. The other Reviewers also raised this issue. We have reorganized related sentences/paragraphs in the revised manuscript.

5. The lower reaches of the Yangtze River flows through the agricultural plain, and the use of a large amount of chemical fertilizers may have a great influence on river nitrogen. Some discussion on this may be necessary.

**Response:** Thank you for pointing this out. In the lower reach of the Changjiang River, the nitrogen and phosphorus concentrations have shown significant increasing trends over recent decades, largely due to the increasing fertilizer usage over the river basin (Li et al., 2007; Yan et al., 2010; Gao et al., 2012). In the revised manuscript, the sentences about the increasing trend of nutrients in recent decades have been added.

References:

Gao, L., Li, D., and Zhang, Y.: Nutrients and particulate organic matter discharged by the Changjiang (Yangtze River): Seasonal variations and temporal trends, J. Geophys. Res.: Biogeo., 117, G04001, https://doi.org/10.1029/2012JG001952, 2012.

Li, M., Xu, K., Watanabe, M., and Chen, Z.: Long-term variations in dissolved silicate, nitrogen, and phosphorus flux from the Yangtze River into the East China Sea and impacts on estuarine ecosystem, Estuar. Coast. Shelf S., 71, 3-12, https://doi.org/10.1016/j.ecss.2006.08.013, 2007.

Yan, W., Mayorga, E., Li, X., Seitzinger, S. P., and Bouwman, A. F.: Increasing anthropogenic nitrogen inputs and riverine DIN exports from the Changjiang River basin under changing human pressures, Global Biogeochem. Cy., 24, GB0A06, https://doi.org/10.1029/2009GB003575, 2010.

Again, we appreciate the Reviewer for the constructive and insightful comments and time spent on our manuscript. The comments have greatly improved our manuscript. We hope that our revised manuscript now meets the standard set by *Biogeosciences*.

---

## Author Comment (AC2)

Response to Comments (paper # EGUsphere-2022-30)

**Variations in dissolved and particulate organic carbon in the lower Changjiang River on time scales from seasonal to decades**

Yue Ming[1], Lei Gao[1]*, and Laodong Guo[2]

1. State Key Laboratory of Estuarine and Coastal Research, East China Normal University, 3663 North Zhongshan Road, Shanghai 200241, China
2. University of Wisconsin-Milwaukee, School of Freshwater Sciences, 600 E Greenfield Ave., Milwaukee, WI 53204, USA

Correspondence to: lgao@sklec.ecnu.edu.cn

We gratefully thank Reviewer for your time and valuable comments on our manuscript. We have carefully considered these comments and revised the manuscript accordingly. Our response to reviewers' comments point by point is given below. The original comments (in blue text) are also provided followed by our detailed response (regular font size).

**Reviewer #2:**
General comment
In this work, Ming and colleagues reported the organic carbon dynamics in the Yangtze (Changjiang) River on seasonal to decadal scales. The authors attempted to evaluate DOC and POC concentrations, fluxes, and chemical composition based on monthly sampling results during the 2016-2020 and earlier results retrieved from the literature. OC transport by large rivers has become an increasingly important research topic due to its complex responses to global change. For the Yangtze River studied in this work, clearly its OC dynamics are not an exception and have been substantially modified by both climate change and human activities. The latter may be the dominant factor in consideration of the large number of dams constructed in the river basin. While this research work fits well with the scope of the journal Biogeosciences, OC transport in the Yangtze River has been widely examined in the literature (see a

few examples in the specific comment above). E.g., more than 30 research articles have studied DOC and/or POC in this river in the recent 10-15 years. I don't think this manuscript represents any significant research breakthroughs in this field/study area.

**Response:** Thank you for this comment. However, we believe that this study has brought quite a lot of new and important information to the reader and is of great significance in the literature. In their comments, the Reviewer listed a number of references to argue that the biogeochemical behaviors of POC and DOC in the Changjiang River have been widely studied and that this study represents no significant breakthrough. However, our study indeed contains new data and results that cannot be found in these previous studies (see references cited in the manuscript and those pointed by this reviewer).

First, in this study, we carried out ultrafiltration to examine the variations and behaviors of DOM with different size-fractions, and to establish their relationships with POC. To the best of our knowledge, this approach has never been used in these previous studies (see references provided by the Reviewer) and, in general, seldom employed in studies in the literature. In fact, our group and our colleagues have focused on the biogeochemistry in the Changjiang River and its estuary for more than ten years (e.g., Gao et al., 2012; Zhao et al., 2021), and we are well aware of what is really new knowledge regarding DOM/POM biogeochemistry in this important area.

In addition, this study has collected the most comprehensive data to elucidate the decadal trends in concentrations and chemical properties of POC and DOC in the study area. The newest data for the most recent ten years are very important because the Changjiang River basin has been experiencing intensive changes during this period. As a matter of fact, new findings presented in this manuscript have never been reported or raised in the literature. For example, we found that DOC concentrations, and values of POC (%), $\delta^{13}C$, and $\delta^{15}N$ all significantly increased ($P < 0.05$) over the recent decades. These decadal trends are new and important findings because they suggested that the Changjiang River system is just transforming from a "primitive" one (more contributed by allochthonous materials) to a "civilized" one (more contributed by autochthonous materials), consistent with the rapid economic and social developments in China.

References:

Gao, L., Li, D., and Zhang, Y.: Nutrients and particulate organic matter discharged by the Changjiang (Yangtze River): Seasonal variations and temporal trends, J. Geophys. Res.: Biogeo., 117, G04001, https://doi.org/10.1029/2012JG001952, 2012.

Zhao, L., Gao, L., and Guo, L.: Seasonal variations in molecular size of chromophoric dissolved organic matter from the lower Changjiang (Yangtze) River, J. Geophys. Res.: Biogeo., 126, e2020JG006160, https://doi.org/10.1029/2020JG006160, 2021.

When the authors attempted to explain the different, sometimes contrasting, responses of DOC and POC to flow and sediment changes, they tried to use 'flushing' and 'dilution' effects. I don't challenge these effects and I do believe they have naturally occurred. However, many statements are speculative and lack of solid evidence to support (see specific comments below), and even contradictory (e.g., L249-254).

**Response:** Thank you for pointing these out. In the revised manuscript, we have revised related text and added relevant references to support our statements. Please see below our response to your specific comments.

Overall, the grammar and syntax are shaky throughout the text. There are numerous grammar errors with many confusing statements. I started correcting this in detail, but was soon overwhelmed by the number of changes that could be made. A further language editing is needed before its resubmission.

**Response:** Thank you for the suggestion. In the revised manuscript, grammar and syntax have been thoroughly polished by an English native speaker.

Specific comments (with line number):
L23: What kinds of changes have been observed in the Yangtze River? Please elaborate.

**Response:** Changes in the Changjiang River included increased nutrient concentrations (Li et al., 2007; Yan et al., 2010; Gao et al., 2012) and sharply decreased SPM concentrations and fluxes owing to the constructions of the tremendous dams and reservoirs during recent decades (Yang et al., 2011). In addition, changes in seasonal variation patterns of sediment fluxes have been reported (Dai et al., 2008, 2016). In this study, one of our main purposes was to elucidate decadal changes of organic carbon (including POC and DOC, as well as their phase partitioning), one of the most important biogeochemical components in the river basin.

We have added these explanations in the revised manuscript. Again, we appreciate all the details comments from this reviewer.

References:

Dai, Z., Du, J., Li, J., Li, W., and Chen, J.: Runoff characteristics of the Changjiang River during 2006: Effect of extreme drought and the impounding of the Three Gorges Dam, Geophys. Res. Lett., 35, L07406, https://doi.org/10.1029/2008GL033456, 2008.

Dai, Z., Fagherazzi, S., Mei, X., and Gao, J.: Decline in suspended sediment concentration delivered by the Changjiang (Yangtze) River into the East China Sea between 1956 and 2013, Geomorphology, 268, 123-132, https://doi.org/10.1016/j.geomorph.2016.06.009, 2016.

Gao, L., Li, D., and Zhang, Y.: Nutrients and particulate organic matter discharged by the Changjiang (Yangtze River): Seasonal variations and temporal trends, J. Geophys. Res.: Biogeo., 117, G04001, https://doi.org/10.1029/2012JG001952, 2012.

Li, M., Xu, K., Watanabe, M., and Chen, Z.: Long-term variations in dissolved silicate, nitrogen, and phosphorus flux from the Yangtze River into the East China Sea and impacts on estuarine ecosystem, Estuar. Coast. Shelf S., 71, 3-12, https://doi.org/10.1016/j.ecss.2006.08.013, 2007.

Yan, W., Mayorga, E., Li, X., Seitzinger, S. P., and Bouwman, A. F.: Increasing anthropogenic nitrogen inputs and riverine DIN exports from the Changjiang River basin under changing human pressures, Global Biogeochem. Cy., 24, GB0A06, https://doi.org/10.1029/2009GB003575, 2010.

Yang, S. L., Milliman, J. D., Li, P., and Xu, K.: 50,000 dams later: Erosion of the Yangtze River and its delta, Global Planet. Change, 75, 14-20, https://doi.org/10.1016/j.gloplacha.2010.09.006, 2011.

L24-26: This statement is quite confusing and unclear. The authors need to reword this to make it directly related to the study river.

**Response:** This sentence, as the first sentence of the Introduction, will be modified in the revised manuscript. "*The changes over the Changjiang River basin included the sharply decreased SPM concentrations and transport fluxes, the increasing nutrient concentrations (Li et al., 2007; Yan et al., 2010; Gao et al., 2012), as well as the changed seasonal variation patterns of water discharge and sediment fluxes (Dai et al., 2008, 2013), with the constructions of the tremendous dams (Yang et al., 2011).*"

L27: change 'should also be' to 'has also been'

**Response:** Corrected. Thank you.

L30: change 'tremendous amount of' to 'large quantities of'

**Response:** Corrected. Thank you.

L44-47: this statement is problematic. For organic matter like OC, there are quite a lot of studies already available in the literature. E.g., Sun et al., 2021. Source, transport and fate of terrestrial organic carbon from Yangtze River during a large flood event: Insights from multiple-isotopes ($\delta^{13}$C, $\delta^{15}$N, $\Delta^{14}$C) and geochemical tracers. Geochimica et Cosmochimica Acta. Shi et al., 2016. The spatial and temporal distribution of dissolved organic carbon exported from three Chinese rivers to the China Sea. Plos One. Wu et al., 2018. Spatiotemporal variation of the quality, origin, and age of particulate organic matter transported by the Yangtze River (Changjiang).JGR Biogeosciences. Also, Wu et al., 2015. Temporal variability of particulate organic carbon in the lower Changjiang (Yangtze River) in the post-Three Gorges Dam period: Links to anthropogenic and climate impacts. JGR Biogeosciences; Yu et al., 2011. Impact of extreme drought and the Three Gorges Dam on transport of particulate terrestrial organic carbon in the Changjiang (Yangtze) River. JGR Earth Surface. Zhang et al., 2014. The spatiotemporal distribution of dissolved inorganic and organic carbon in the main stem of the Changjiang (Yangtze) River and the effect of the Three Gorges Reservoir. JGR Biogeosciences.

**Response:** Thank you for providing all the important references, some of these references have been cited in our original manuscript. Notice that the first reference, Sun et al. (2021), mainly focused on the DOM behaviors in the Changjiang River Estuary, which is obviously different from our study area. The remaining five references, Shi et al. (2016), Wu et al. (2015, 2018), Yu et al. (2011), Zhang et al. (2014), all reported data prior to 2011, and their contents are also quite different from ours. In contrast, our manuscript reported the most updated data and new findings.

Most importantly, this study, using a statistical method, proved that significant decadal trends really existed for several important biogeochemical parameters, such as DOC concentrations, POC (%), and $\delta^{13}$C, which have not been identified in all the previous studies (see above references), but can be linked to recent rapid changes due to climate and environmental changes in the basin. Similarly, these decadal trends in organic carbon species in the Changjiang River have not been reported in the literature. As pointed out by Reviewer #1, this manuscript really provided new and important information to the literature.

References:

Sun, X., Fan, D., Cheng, P., Hu, L., Sun, X., Guo, Z., and Yang, Z.: Source, transport and fate of terrestrial organic carbon from Yangtze River during a large flood event: Insights from multiple-isotopes ($\delta^{13}$C, $\delta^{15}$N, $\Delta^{14}$C) and geochemical tracers, Geochim. Cosmochim. Ac., 308, 217-236, https://doi.org/10.1016/j.gca.2021.06.004, 2021.

Shi, G., Peng, C., Wang, M., Shi, S., Yang, Y., Chu, J., Zhang, J., Lin, G., Shen, Y., and Zhu, Q.: The spatial and temporal distribution of dissolved organic carbon exported from three Chinese rivers to the China Sea, PloS one, 11, e0165039, https://doi.org/10.1371/journal.pone.0165039, 2016.

Wu, Y., Eglinton, T. I., Zhang, J., and Montlucon, D. B.: Spatiotemporal variation of the quality, origin, and age of particulate organic matter transported by the Yangtze River (Changjiang), J. Geophys. Res.: Biogeo., 123, 2908-2921, https://doi.org/10.1029/2017JG004285, 2018.

Wu, Y., Bao, H., Yu, H., Zhang, J., and Kattner, G.: Temporal variability of particulate organic carbon in the lower Changjiang (Yangtze River) in the post‐Three Gorges Dam period: Links to anthropogenic and climate impacts, J. Geophys. Res.: Biogeo., 120, 2194-2211, https://doi.org/10.1002/2015JG002927, 2015.

Yu, H., Wu, Y., Zhang, J., Deng, B., and Zhu, Z.: Impact of extreme drought and the Three Gorges Dam on transport of particulate terrestrial organic carbon in the Changjiang (Yangtze) River, J. Geophys. Res.: Earth, 116, F04029, https://doi.org/10.1029/2011JF002012, 2011.

Zhang, L., Xue, M., Wang, M., Cai, W. J., Wang, L., and Yu, Z.: The spatiotemporal distribution of dissolved inorganic and organic carbon in the main stem of the Changjiang (Yangtze) River and the effect of the Three Gorges Reservoir, J. Geophys. Res.: Biogeo., 119, 741-757, https://doi.org/10.1002/2012JG002230, 2014.

L64: what is a position well? please provide more details.

**Response:** Thank you for this comment. The Xuliujing station is the lowest freshwater end-member station well representing the overall Changjiang river discharge and fluxes. Below this station, the Changjiang River begins to bifurcate into the South and North Branches before finally discharging into its estuary and the East China Sea. Thus, in the literature, the data collected at the Xuliujing station were always selected as those representing the overall fluxes from the entire river basin to the East China Sea. We have added related sentences into the revised manuscript.

L75: while the OC was sampled at Xuliujing station, flow and sediment data were measured at Datong station (500 km apart), how were the flow and sediment inputs

between the two stations accounted for, especially there is a large lake (fig. 1) in between the two sites.

**Response:** Thank you for pointing this out. Reviewer #3 also raised a similar question. Although the two stations between Datong (for discharges) and Xuliujing (for sampling) are far apart, however, the total discharges from the several small tributaries (including those from the large lake you mentioned) between Datong and Xuliujing only accounted for 1.2% of the Datong's annual discharge (Mei et al., 2019).

The large lake, Lake Taihu, located between the two stations has relatively longer retention times, i.e., 210 d, on average, over 1986–2006 and 184 d over 2007–2018 (Zhu et al., 2021). Both its input (i.e., $80.9 \times 10^8$ m$^3$/yr in 1986–2006 and $111.6 \times 10^8$ m$^3$/yr in 2007–2018) and output flow ($88.9 \times 10^8$ m$^3$/yr in 1986–2006 and $104.8 \times 10^8$ m$^3$/yr in 2007–2018) were very small compared to the discharge measured at Datong station (average $8931 \times 10^8$ m$^3$/yr over 1950–2015, according to CWRC [2015]). Therefore, this lake should have a small influence on the discharge values measured in the Changjiang River mainstream. We have added this information to the revised manuscript.

References:

CWRC (Changjiang Water Resources Commission) (Eds.): Changjiang Sediment Bulletin 2015, Changjiang Press, Wuhan, 2016.

Mei, X., Zhang, M., Dai, Z., Wei, W., and Li, W.: Large addition of freshwater to the tidal reaches of the Yangtze (Changjiang) River, Estuar. Coast., 42, 629-640, https://doi.org/10.1007/s12237-019-00518-0, 2019.

Zhu, W., Cheng, L., Xue, Z., Feng, G., Wang, R., Zhang, Y., Zhao, S., and Hu, S.: Changes of water exchange cycle in Lake Taihu (1986–2018) and its effect on the spatial pattern of water quality (in Chinese), Journal of Lake Sciences, 33, 1087-1099, http://doi.org/10.1837/2021.0411, 2021.

L115: I don't think this a good reference to support the DOC measurement method.

**Response:** This reference has been replaced by Guo and Santschi (1997) in the revised manuscript.

Reference:

Guo, L. and Santschi, P.: Measurements of dissolved organic carbon (DOC) in sea water by high temperature combustion method, Acta Oceanol. Sin., 16, 339-353, 1997.

L121: for SPM, have the filters been weighed before sampling? If not, this may cause errors as each filter weight is different.

**Response:** Thank you for the specific comment. The pre-combusted filters had been weighed before sampling. Once the water samples were collected, they were filtered, as soon as possible, through pre-weighed Whatman GF/F filters (with a pore size of 0.7 μm) that had been pre-combusted (450°C, 5 h) and pre-weighed ($W_1$) before the sample collectetion. After filtration in the laboratory, the filters were immediately frozen at -20°C until further analysis.

In the laboratory, the filter samples were oven-dried (50°C, 48 h) and weighed again ($W_2$). The weight differences between the dried filters and their blank weights measured before the cruises ($W_2 - W_1$) were divided by the water volume of each sample to calculate the SPM mass concentration (in mg/L, referred to as SPM concentrations hereafter). We have added these technical details in the revised manuscript.

L133: how were the decadal data collected. No details were provided. Also, if they were collected from the literature as indicated later in the text, have the authors checked the data quality and sampling consistency. E.g., are the sampling results at different sites (e.g., Datong and Xuliujing) comparable? How to resolve the lake impact? These should have been clearly provided in the text.

**Response:** The details about where these data were collected and the related references had been provided in related figure captions. Regarding possible differences in data between Datong and Xuliujing stations, although the two stations between Datong (for discharges) and Xuliujing (for sampling) are far apart, the total discharges from the several small tributaries between Datong and Xuliujing only accounted for 1.2% of the Datong's annual discharge (Mei et al., 2019). Therefore, the discharge measured at Datong has always been used to represent the ultimate discharge from the Changjiang River. In terms of the chemical properties, the results

from Liu et al. (2003) suggested that the concentrations of all the nutrients ($NO_3^-$, $SiO_3^{2-}$, $PO_4^{3-}$, $NH_4^+$, and DIN) were relatively constant over the whole lower reach of the Changjiang River (please see Figure L2, the letter L means the reference of Liu et al. [2003], similarly hereafter).

[Figure]

Figure L2. Concentrations of nutrients in the main stream of the Changjiang, which are plotted against the distance from the river mouth.

Bao et al. (2015) carried out two samplings (in October 2009 and July-August 2010) in the middle and lower Changjiang mainstream, and they showed that DOC

concentrations between Xuliujing and the nearest upward stations (with distances even longer than that between Datong and Xuliujing) were generally 0.1 mg-C/L (equal to 8.3 μM). Wang et al. (2019) also showed that the DOC and CDOM (quantified by $a_{254}$) over the distance from Xuliujing to stations about 500 km upward were also relatively stable, especially considering the variation ranges after data from the middle and lower reaches had all been included. Similarly, Wu et al. (2018) carried out two sampling cruises in October 2009 and August 2010, and they observed that POC (%), $\delta^{13}C$, and conventional ages were rather stable in the SPM samples over the whole lower reach (see their Table W2). Yu et al. (2011) conducted samplings in April-May 2011 and October-November 2006 in the middle and lower reaches of the Changjiang River, and they also found that POC (%) (0.92 ± 0.06 in 2003, and 1.73 ± 0.23 in 2006) and $\delta^{13}C$ (‰) values (–24.90‰ ± 0.14‰ in 2003, and –24.90‰ ± 0.06‰ in 2006) were rather similar over the whole lower reach (see their Table Y2). Data from our group also suggested that the chemical properties between Xuliujing and Wuhu (close to Datong) in May 2021 were quite similar (Table R1 below).

Table W2. TSM concentrations (TSM samples only), organic carbon contents (POC%), and bulk stable carbon isotope ($\delta^{13}C$, ‰) and radiocarbon compositions (conventional $^{14}C$ age, years BP) of organic carbon in suspended particulate matter and sediments in the lower reaches of the Changjiang river.

| Sampling period | Station | Water depth (m) | TSM (mg/L) | POC% | $\delta^{13}C$ | Conventional age (BP) |
|---|---|---|---|---|---|---|
| Oct 2009 | Wuhu | 0 | 69.0 | 1.0 | -25.7 | 2190 |
| | | 4.5 | 57.3 | 0.9 | -25.7 | 2200 |
| | | 8 | 61.0 | 0.9 | -25.5 | 1900 |
| | | 13.5 | 55.5 | 0.8 | -25.9 | 2360 |
| | Xuliujing | 0 | 27.2 | 0.8 | -26.0 | 2090 |
| | | 4.5 | 33.7 | 0.8 | -25.4 | 2350 |

| | | | | | | |
|---|---|---|---|---|---|---|
| | | 9 | 36.0 | 0.8 | -25.7 | 2250 |
| | | 14.5 | 40.7 | 0.8 | -25.6 | 2240 |
| Aug 2010 | Jiujiang | 0 | 157.3 | 1.0 | -25.6 | 1570 |
| | | 18 | 173.7 | 1.2 | -25.1 | |
| | Jiangyin | 0 | 136.5 | 1.2 | -25.1 | 2960 |

Table R1. Comparison of chemical properties between Xuliujing and Wuhu stations in the Changjiang River mainstream, collected in May 2021 (unpublished data from our group).

| Station | Distance (km) | SPM (mg/L) | POC (μmol/L) | DOC (μmol/L) | POC (%) | $\delta^{13}C$ (‰) | $\delta^{15}N$ (‰) | POC/PN (mol/mol) |
|---|---|---|---|---|---|---|---|---|
| Wuhu | 492 | 33.7 | 73 | 155 | 2.6 | -23.9 | 5.7 | 7.7 |
| Xuliujing | 114 | 40.1 | 80 | 152 | 2.4 | -23.8 | 6.1 | 7.0 |

In the revised manuscript, we have added some sentences into the revised manuscript, explaining that the samples collected between Datong and Xuliujing were generally similar in chemical properties or no systematic differences/changes could be found between samples collected at these two stations.

References:

Bao, H., Wu, Y., and Zhang, J.: Spatial and temporal variation of dissolved organic matter in the Changjiang: fluvial transport and flux estimation, J. Geophys. Res.: Biogeo., 120, 1870-1886, https://doi.org/10.1002/2015JG002948, 2015.

Liu, S. M., Zhang, J., Chen, H. T., Wu, Y., Xiong, H., and Zhang, Z. F.: Nutrients in the Changjiang and its tributaries, Biogeochemistry, 62, 1-18, https://doi.org/10.1023/A:1021162214304, 2003.

Mei, X., Zhang, M., Dai, Z., Wei, W., and Li, W.: Large addition of freshwater to the tidal reaches of the Yangtze (Changjiang) River, Estuar. Coast., 42, 629-640, https://doi.org/10.1007/s12237-019-00518-0, 2019.

Wang, X., Wu, Y., Bao, H., Gan, S., and Zhang, J.: Sources, transport, and transformation of dissolved organic matter in a large river system: Illustrated by the Changjiang River, China, J. Geophys. Res.: Biogeo., 124, 3881-3901, https://doi.org/10.1029/2018JG004986, 2019.

Wu, Y., Eglinton, T. I., Zhang, J., and Montlucon, D. B.: Spatiotemporal variation of the quality, origin, and age of particulate organic matter transported by the Yangtze River (Changjiang), J. Geophys. Res.: Biogeo., 123, 2908-2921, https://doi.org/10.1029/2017JG004285, 2018.

Yu, H., Wu, Y., Zhang, J., Deng, B., and Zhu, Z.: Impact of extreme drought and the Three Gorges Dam on transport of particulate terrestrial organic carbon in the Changjiang (Yangtze) River, J. Geophys. Res.: Earth, 116, F04029, https://doi.org/10.1029/2011JF002012, 2011.

About the effect of the lakes, according to CWRC and taking data from 2019 as an example, the outflows from Lakes Doting and Poyang to the Changjiang were 2.87 $\times$ 10$^{11}$ m$^3$/yr and 1.94 $\times$ 10$^{11}$ m$^3$/yr, which accounted for about 30.8% and 20.8%, respectively, to the discharge at Datong station (9.33 $\times$ 10$^{11}$ m$^3$/yr). This conclusion is consistent with those suggested by Bao et al. (2015). At the same time, however, it must be noted that the inflows to the two lakes (2.52 $\times$ 10$^{11}$ m$^3$/yr and 1.57 $\times$ 10$^{11}$ m$^3$/yr, respectively) were also very large, which largely reflected the contributions from large tributaries, such as the Xiangjiang River (with discharge of 0.9 $\times$ 10$^{11}$ m$^3$/yr) to Lake Dongting and the Ganjiang River (with discharge of 1.0 $\times$ 10$^{11}$ m$^3$/yr) to Lake Poyang.

Wu et al. (2014) suggested that the water retention time of Lake Poyang was relatively low, approximately 10 d, a value that was considerably less than the other two large freshwater lakes in China (i.e., Lake Taihu and Lake Chaohu, with retention times of 264 and 127 d, respectively). Furthermore, the water retention time varied among seasons in Lake Poyang, with low values in the dry (2.7 d) and mid-dry (12.5 d) seasons but a comparatively high value in the wet season (25.5 d). Pan et al. (2009) also suggested that the retention times for Lakes Dongting and Poyang were 18.2 and 10.0 d, respectively. Liu et al. (2016) suggested that the average retention time was less than 10 d along the main flow channels of Lake Poyang; whereas approximately 30 d was estimated in the summer. The short retention times of the two lakes suggested that they were more like a passageway of tributaries rather than a reaction vessel.

The chemical properties in these lake waters were also highly variable and largely depended on the specific locations and the retention times where samples were

collected. According to Bao et al. (2014), the chemical parameters (POC (%), PN (%), POC/PN ratio, and $\delta^{13}C$) measured in the two lakes were generally similar to, or within the variation ranges of those measured at stations in the nearby Changjiang River mainstream during the same sampling time period.

Furthermore, the influence of lakes on the water chemistry in the Changjiang River mainstream is very important and complex. Based on data collected during our field sampling in May 2021, the chemical properties did not have large differences between lake waters and the Changjiang River mainstream (see also Table R2), including DOC concentration, POC (%), $\delta^{13}C$, $\delta^{15}N$, and POC/PN.

Table R2. Comparisons in chemical properties of Lakes Dongting and Poyang and the Changjing River at the Xuliujing station during May 2021 (unpublished data from our group).

| Lake or Station | Distance (km) | SPM (mg/L) | POC (μmol/L) | DOC (μmol/L) | POC (%) | $\delta^{13}C$ (‰) | $\delta^{15}N$ (‰) | POC/PN (mol/mol) |
|---|---|---|---|---|---|---|---|---|
| Dongting | 1361 | 21 | 126 | 171 | 7.2 | -26.1 | 5.2 | 6.6 |
| Poyang | 840 | 45 | 169 | 141 | 4.5 | -25.0 | 4.7 | 7.9 |
| Xuliujing | 114 | 40 | 80 | 152 | 2.4 | -23.8 | 6.1 | 6.0 |

As suggested by Reviewer #2, the influence of lakes is very important, and we have added these discussions into the revised manuscript.

References:

Bao, H., Wu, Y., and Zhang, J.: Spatial and temporal variation of dissolved organic matter in the Changjiang: fluvial transport and flux estimation, J. Geophys. Res.: Biogeo., 120, 1870-1886, https://doi.org/10.1002/2015JG002948, 2015.

Bao, H., Wu, Y., Zhang, J., Deng, B., and He, Q.: Composition and flux of suspended organic matter in the middle and lower reaches of the Changjiang (Yangtze River)—impact of the Three Gorges Dam and the role of tributaries and channel erosion, Hydrol. Process., 28, 1137-1147, https://doi.org/10.1002/hyp.9651, 2014.

Liu, X., Li, Y.-L., Liu, B.-G., Qian, K.-M., Chen, Y.-W., and Gao, J.-F.: Cyanobacteria in the complex river-connected Poyang Lake: horizontal distribution and transport, Hydrobiologia, 768, 95-110, https://doi.org/10.1007/s10750-015-2536-2, 2016.

Pan, B.-Z., Wang, H.-J., Liang, X.-M., and Wang, H.-Z.: Factors influencing chlorophyll *a* concentration in the Yangtze-connected lakes, Fresen. Environ. Bull., 18, 1894-1900, 2009.

Wu, Z., Lai, X., Zhang, L., Cai, Y., and Chen, Y.: Phytoplankton chlorophyll *a* in Lake Poyang and its tributaries during dry, mid-dry and wet seasons: a 4-year study, Knowl. Manag. Aquat. Ec., 06, https://doi.org/10.1051/kmae/2013088, 2014.

L142: the authors mixed results and discussion together in this section, but also presented a separate 'discussion' section. The structure is quite confusing and different to follow. I would suggest the authors move all discussion words into the 'discussion' section and keep only descriptive results here.

**Response:** Thank you for this comment. Reviewer #1 also raised the similar comment.

We have re-organized our manuscript and moved all the discussion sentences into the

Discussion Section.

L147: change 'should' to 'may' or 'might'

**Response:** Corrected. Thank you.

L158-161: this is a speculative statement. Do the authors have evidence to support this?

**Response:** As we have explained earlier, due to the decreased deforestation and

increasing dam trapping, the SPM concentrations have had much weaker correlations

with river discharge over the past two decades. In the revised manuscript, we have

modified this sentence and cited Dai et al. (2016) to support our statement. Please also

see our response to your next comment.

Reference:

Dai, Z., Fagherazzi, S., Mei, X., and Gao, J.: Decline in suspended sediment concentration delivered by the Changjiang (Yangtze) River into the East China Sea between 1956 and 2013, Geomorphology, 268, 123-132, https://doi.org/10.1016/j.geomorph.2016.06.009, 2016.

L192-194: as the lower Yangtze have been heavily regulated by the Three Gorge Dam, does this distinct seasonal pattern reflect the impact of the dam? Flood seasons with

high sediment concentrations (reduced light penetration and turbulent flow conditions) are typically not favourable for autochthonous production.

**Response:** Excellent comment. In the Changjiang River, it is generally regarded that higher river discharge during summer should also be accompanied by higher SPM concentrations, largely due to the enhanced soil erosion caused by deforestation. However, due to the increasing intensive dam trapping and decreased deforestation, the difference in SPM concentration between flood and dry seasons and thus the correlation between SPM and discharge became less obvious or almost disappeared (please see Figures D11 and D12 of Dai et al. [2016]). Based on our own data (Figure G4a, and those in Gao et al. [2012]), no clear trends or significant differences in SPM concentrations were observed between flood seasons and dry seasons in the Changjiang River.

[Figure]

Figure D11. Daily SSC (suspended sediment concentration) and water discharge in different flood years: A) 1964; (B) 1973; (C) 1983; (D) 1998; (E) 2010.

[Figure]

Figure D12. Daily SSC and water discharge during different drought years: (A) 1963; (B) 1978; (C) 2011.

[Figure]

Figure G4a. Variations of the monthly average values (±standard deviation) of SPM abundance measured at station #4 during the sampling period September 2009 to August 2010.

In fact, our results from these studies using parameters like POC (%), POC/PN, $\delta^{13}C$, and $\delta^{15}N$, as well as those from the literature using other parameters (e.g., n-alkane data in Zhao et al. [2022]) all suggested that organic carbon pools in the Changjiang River always had more autochthonous sources in summer rather than in winter. A similar conclusion has also been drawn in the nearby Yellow River (Tao et

al., 2018). As suggested by Reviewer #2, we have added these explanations into the revised manuscript.

References:

Dai, Z., Fagherazzi, S., Mei, X., and Gao, J.: Decline in suspended sediment concentration delivered by the Changjiang (Yangtze) River into the East China Sea between 1956 and 2013, Geomorphology, 268, 123-132, https://doi.org/10.1016/j.geomorph.2016.06.009, 2016.

Gao, L., Li, D., and Zhang, Y.: Nutrients and particulate organic matter discharged by the Changjiang (Yangtze River): Seasonal variations and temporal trends, J. Geophys. Res.: Biogeo., 117, G04001, https://doi.org/10.1029/2012JG001952, 2012.

Tao, S., Eglinton, T. I., Zhang, L., Yi, Z., Montluçon, D. B., McIntyre, C., Yu, M., and Zhao, M.: Temporal variability in composition and fluxes of Yellow River particulate organic matter, Limnol. Oceanogr., 63, S119-S141, https://doi.org/10.1002/lno.10727, 2018.

Zhao, M., Sun, H., Liu, Z., Bao, Q., Chen, B., Yang, M., Yan, H., Li, D., He, H., Wei, Y., and Cai, G.: Organic carbon source tracing and the BCP effect in the Yangtze River and the Yellow River: Insights from hydrochemistry, carbon isotope, and lipid biomarker analyses, Sci. Total Environ., 812, 152429, https://doi.org/10.1016/j.scitotenv.2021.152429, 2022.

L241: any references to support this?

**Response:** Thank you for the comment. References for other world rivers, such as the Mississippi River (Cai et al., 2015) and the Yukon River (Guo et al., 2012) showing that DOC concentrations were generally positively correlated with river discharge, are provided in the revised manuscript. This sentence has also been modified.

References:

Cai, Y., Guo, L., Wang, X., and Aiken, G.: Abundance, stable isotopic composition, and export fluxes of DOC, POC, and DIC from the Lower Mississippi River during 2006–2008, J. Geophys. Res.: Biogeo., 120, 2273-2288, https://doi.org/10.1002/2015JG003139, 2015.

Guo, L., Cai, Y., Belzile, C., and Macdonald, R. W.: Sources and export fluxes of inorganic and organic carbon and nutrient species from the seasonally ice-covered Yukon River, Biogeochemistry, 107, 187-206, https://doi.org/10.1007/s10533-010-9545-z, 2012.

L249-254: the two sentences are contradictory to each other.

**Response:** Thank you very much for pointing this out. In the revised manuscript, these sentences have been modified. "*On one hand, there is a significant correlation*

*in δ¹³C values between the HMW-DOC and bulk POC pools (Figure 5b), suggesting that the two organic carbon pools had similar background sources, or were closely related. On the other hand, regardless of flood or dry seasons, the δ¹³C values measured in the HMW-DOC were generally slightly lower than those in the bulk POC, suggesting that the HMW-DOC seemed to have more terrestrial signals compared to POC. In other words, the POC pool contained more autochthonous components than the HMW-DOC (Gao et al., 2014; Wang et al., 2021b).*"

L265-268: again, these statements are confusing. The authors may also need to pay attention to the upstream dam impacts on flow regulation and sediment flushing.

**Response:** Good comment. Following your suggestion, this sentence has been modified. It now reads: "*Probably owing to the upstream dam impacts on flow regulation and sediment flushing, during the extreme dry season between November 2019 and January 2020, POC concentrations (Figure 2b) and δ¹³C in POC and HMW-DOC (Figures 3b and 5b) consistently showed their lowest values.*"

L297: this is not a good reference to support this statement. Also, this paragraph is quite general.

**Response:** We agree. This statement has been removed from the revised manuscript.

L308-311: again, a speculative statement. Do the authors have evidence to support this argument?

**Response:** Thank you for the comment and we agree. Reviewer #3 also raised the similar comment. To avoid speculative statement, this sentence has been removed from the revised manuscript.

L323: why not include 1-10 kDa DOC in fig 7?

**Response:** By using ultrafiltration, the bulk DOC pools can be separated into the four size fractions, including the < 1 kDa, 1–3 kDa, 3–10 kDa, and 10 kDa–0.7 μm. Here, we only compared the parameters for size 10 kDa–0.7 μm with those of POC, simply because the DOC with size > 10 kDa is the most similar fraction with POC in terms of

sizes. Thus, it should be most meaningful and most possible to establish some significant or strong relationships between these two OM fractions.

L324: note that you have not yet discussed SPM and POC concentrations until now. The structure here is confusing and difficult to follow.

**Response:** Thank you for pointing this out. We have added related sentences and discussion about the relationship between SPM and POC concentrations in the revised manuscript.

L333-346: you cannot simply use studies in other rivers to support your arguments in Yangtze.

**Response:** We agree. In fact, one of the cited references, Wang et al. (2016), in the original manuscript, had a focus on the Changjiang River. In the revised manuscript, we have added another important reference, Shan et al. (2020), to further support the conclusion that the $\delta^{13}C$ values in DIC were generally higher than the values in DOC in the Changjiang River.

References:

Shan, S., Qi, Y., Luo, C., Fu, W., Xue, Y., and Wang, X.: Carbon isotopic constrains on the sources and controls of the terrestrial carbon transported in the four large rivers in China, Advances in Earth Science, 35, 948-961, http://doi.org/10.11867/j.issn.1001-8166.2020.078, 2020.

Wang, X., Luo, C., Ge, T., Xu, C., and Xue, Y.: Controls on the sources and cycling of dissolved inorganic carbon in the Changjiang and Huanghe River estuaries, China: $^{14}C$ and $^{13}C$ studies, Limnol. Oceanogr., 61, 1358-1374, https://doi.org/10.1002/lno.10301, 2016.

L355: for the annual fluxes, how were them calculated? What is the uncertainty?

**Response:** Good comment. For the monthly fluxes, we generally multiplied the concentration we measured each month, by the monthly average discharge at Datong reported by CWRC. For the annual fluxes, we generally summed the 12 consecutive monthly fluxes. In terms of uncertainties, our previous published paper, Gao et al. (2012), had discussed the potential errors using all the five methods proposed by Webb et al. (2000), and the standard deviation was about 3% for the annual fluxes of POC. We have added the specific uncertainties in the revised manuscript.

References:

Gao, L., Li, D., and Zhang, Y.: Nutrients and particulate organic matter discharged by the Changjiang (Yangtze River): Seasonal variations and temporal trends, J. Geophys. Res.: Biogeo., 117, G04001, https://doi.org/10.1029/2012JG001952, 2012.

Webb, B. W., Phillips, J. M., and Walling, D. E.: A new approach to deriving 'best-estimate' chemical fluxes for rivers draining the LOIS study area, Sci. Total Environ., 251-252, 45-54, https://doi.org/10.1016/S0048-9697(00)00413-7, 2000.

L360: are the authors sure that the finer SPM will be preferentially trapped, not the coarser ones? In my opinion, the increase grain size is more likely caused by the river-bed armouring process after the TGD dam operation.

**Response:** Thank you very much for this valuable comment. In the revised manuscript, we have revised our statements and cited related references. We agree with you that the coarsening of SPM measured in the lower Changjiang River should be mainly caused by the river-bed armoring processes after the TGD operation. After the dam construction, the downstream channel erosion is always found, because the released water is hungry for SPM load. This erosion process winnows the fine-grained particles and causes coarsening of the channel sediments. The coarsening of the bed sediment, particularly over the first few hundred kilometers downstream of the TGD and in the middle reach of the Changjiang River, is likely to reflect the riverbed erosion (Luo et al., 2012). The increasing coarsening of SPM in the lower reach measured at the Datong station is believed to have resulted from the addition of coarser sediment eroded from the riverbed in the middle reach of the Changjiang River (Yang et al., 2011; Luo et al., 2012). We have added this explanation in the revised manuscript.

References:

Luo, X. X., Yang, S. L., and Zhang, J.: The impact of the Three Gorges Dam on the downstream distribution and texture of sediments along the middle and lower Yangtze River (Changjiang) and its estuary, and subsequent sediment dispersal in the East China Sea, Geomorphology, 179, 126-140, https://doi.org/10.1016/j.geomorph.2012.05.034, 2012.

Yang, S. L., Milliman, J. D., Li, P., and Xu, K.: 50,000 dams later: Erosion of the Yangtze River and its delta, Global Planet. Change, 75, 14-20, https://doi.org/10.1016/j.gloplacha.2010.09.006,

2011.

**Response:** In the revised manuscript, we have revised this sentence to read: "*During the same period, the seasonal Mann-Kendall test verified that DOC concentrations had an increasing trend with a much higher slope value (+0.21 μmol/(L·yr), Figure 10a) compared to POC (+0.02 μmol/(L·yr), Figure 9a).*"

L375: The POC% is increasing with increasing sediment particle size. This is in contrast to our common understanding that POC% will usually decrease with increasing sediment particles.

**Response:** Thank you for pointing this out. We should have explained this clearly. Data from CWRC suggested that the gain size of transport SPM had increased over the recent decades. These SPM particles generally meant those background mineral particles because these samples were digested by $H_2O_2$ during the pre-treating protocols before the size measurements, and during this process, the organic components were largely removed. Therefore, the increasing POC (%) values observed in recent decades are not in contrast to the increasing grain size of background mineral SPM reported by CWRC. POC sources also change with large mineral particles trapped by dam and SPM are nowadays mostly biogenic in source, leading to an increase in POC with increasing particle size. We have added these explanations into the revised manuscript. Thank you again for this valuable comment.

L378-380: speculative statement without supporting evidence.

**Response:** Thank you for pointing this out. We have revised this statement, lowering the tone a bit. It now reads: "*This reference value is very important, suggesting that SPM regimes in the Changjiang River probably have completely changed over the past decades. The increased POC (%) contents in the SPM samples attested to an increasing contribution from autochthonous materials to the bulk SPM pool.*

Fig 7: why not include 1-10 kDa DOC as the authors defined 'HMW-DOC' in the text?

**Response:** We should have defined the HMW-DOC more clearly. In the literature, HMW-DOC or colloidal DOC is generally referred to those DOC molecules with sizes > 1 kDa although this is highly operationally defined and depended on specific research (Guo and Santschi, 2007). In the revised manuscript, we have explicitly defined the HMW-DOC in the Method section based on the ultrafiltration membrane we used.

Reference:

Guo, L. and Santschi, P. H.: Ultrafiltration and its applications to sampling and characterisation of aquatic colloids, in: Environmental Colloids and Particles, IUPAC Series on Analytical and Physical Chemistry of Environmental Systems, edited by: Wilkinson, K. J., and Lead, J. R., John Wiley, 159-221, https://doi.org/10.1002/9780470024539.ch4, 2007.

Fig 10a, if the DOC data before 2003 were considered, the temporal trend will be reversed.

**Response:** In Figure 10, the DOC data before 2003 were from two studies in the literature, i.e., the pink point was from Wu et al. (2007) measured at Nantong in May 1997, and the remaining five points were from Duan et al. (2008) measured at Datong during 1998–1999. If only the one data point from Wu et al. (2007) was incorporated into the statistical analysis and those from Duan et al. (2008) were not, an even stronger increasing trend of DOC concentrations over the past several decadal would be found, which is consistent with our conclusion. However, if the data points from Duan et al. (2008) were also incorporated, a reverse trend, as the Reviewer suggested, would be obtained. Here, we are not certain which data are more representative, but it pointed out that more environmentally consistent data are needed to draw a more conclusive trend for DOC in the Changjiang River over the last decades. In the present study, we presented these literature data in Figure 10, but did not incorporate them into the statistical analysis. In the revised manuscript, we have added this explanation into the revised manuscript.

References:

Duan, S., Liang, T., Zhang, S., Wang, L., Zhang, X., and Chen, X.: Seasonal changes in nitrogen and phosphorus transport in the lower Changjiang River before the construction of the Three Gorges Dam, Estuar. Coast. Shelf S., 79, 239-250, https://doi.org/10.1016/j.ecss.2008.04.002, 2008.

Wu, Y., Zhang, J., Liu, S. M., Zhang, Z. F., Yao, Q. Z., Hong, G. H., and Cooper, L.: Sources and distribution of carbon within the Yangtze River system, Estuar. Coast. Shelf S., 71, 13-25, https://doi.org/10.1016/j.ecss.2006.08.016, 2007.

Again, we appreciate the Reviewer for the constructive and insightful comments and time spent on our manuscript. The comments have greatly improved our manuscript. We hope that our revised manuscript now meets the standard set by *Biogeosciences*.

---

## Author Comment (AC3)

Response to Comments (paper # EGUsphere-2022-30)

**Variations in dissolved and particulate organic carbon in the lower Changjiang River on time scales from seasonal to decades**

Yue Ming[1], Lei Gao[1]\*, and Laodong Guo[2]

1. State Key Laboratory of Estuarine and Coastal Research, East China Normal University, 3663 North Zhongshan Road, Shanghai 200241, China
2. University of Wisconsin-Milwaukee, School of Freshwater Sciences, 600 E Greenfield Ave., Milwaukee, WI 53204, USA

Correspondence to: lgao@sklec.ecnu.edu.cn

We gratefully thank Reviewer for your time and valuable comments on our manuscript. We have carefully considered these comments and revised the manuscript accordingly. Our response to reviewers' comments point by point is given below. The original comments (in blue text) are also provided followed by our detailed response (regular font size).

**Reviewer #3:**
Major comments:
This paper presents DOC and POC data at the Xuliujing Station of the Yangtze River over three years. These data are very valuable to explore the seasonal and long-term variations and controls of organic carbon exported by the Yangtze to coastal oceans. However, the analysis is not complete or thorough. Some of the major conclusions are not supported by the presented data.

**Response:** We appreciate the positive comments and suggestions from Reviewer #3.

(1) The authors conclude that higher $^{13}$C-POC in summer was due to autochthonous production in upstream intensified by human activities (e.g., the Three Gorges Dam). However, I am not convinced on this point. The authors should also consider other

factors potentially affecting $^{13}$C signals of organic carbon. First, Poyang Lake and Dongting Lake are very important sources of water to the main channel of Yangtze (20-30%). These large lakes also contribute organic carbon to the main channel.

**Response:** We totally agree and thank you for all the comments. According to CWRC and taking data from 2019 as an example, the outflows from Lakes Doting and Poyang to the Changjiang were $2.87 \times 10^{11}$ m$^3$/yr and $1.94 \times 10^{11}$ m$^3$/yr, which accounted for about 30.8% and 20.8%, respectively, to the discharge at Datong station $(9.33 \times 10^{11}$ m$^3$/yr). This conclusion is consistent with those suggested by Bao et al. (2015). At the same time, however, it must be noted that the inflows to the two lakes $(2.52 \times 10^{11}$ m$^3$/yr and $1.57 \times 10^{11}$ m$^3$/yr, respectively) were also very large, which largely reflected the contributions from large tributaries, such as the Xiangjiang River (with discharge of $0.9 \times 10^{11}$ m$^3$/yr) to Lake Dongting and the Ganjiang River (with discharge of $1.0 \times 10^{11}$ m$^3$/yr) to Lake Poyang.

Wu et al. (2014) suggested that the water retention time of Lake Poyang was relatively low, approximately 10 d, a value that was considerably less than the other two large freshwater lakes in China (i.e., Lake Taihu and Lake Chaohu, with retention times of 264 and 127 d, respectively). Furthermore, the water retention time varied among seasons in Lake Poyang, with low values in the dry (2.7 d) and mid-dry (12.5 d) seasons but a comparatively high value in the wet season (25.5 d). Pan et al. (2009) also suggested that the retention times for Lakes Dongting and Poyang were 18.2 and 10.0 d, respectively. Liu et al. (2016) suggested that the average retention time was less than 10 d along the main flow channels of Lake Poyang; whereas approximately 30 d was estimated in the summer. The short retention times of the two lakes suggested that they were more like a passageway of tributaries rather than a reaction vessel.

The chemical properties in these lake waters were also highly variable and largely depended on the specific locations and the retention times where samples were collected. According to Bao et al. (2014), the chemical parameters (POC (%), PN (%), POC/PN ratio, and δ$^{13}$C) measured in the two lakes were generally similar to, or within the variation ranges of those measured at stations in the nearby Changjiang River mainstream during the same sampling time period.

Furthermore, the influence of lakes on the water chemistry in the Changjiang River mainstream is very important and complex. Based on data collected during our field sampling in May 2021, the chemical properties showed some differences, although not large, between lake waters and the Changjiang River mainstream (see also Table R1), including DOC concentration, POC (%), $\delta^{13}$C, $\delta^{15}$N, and POC/PN.

Table R1. Comparisons in chemical properties of Lakes Dongting and Poyang and the Changjing River at the Xuliujing station during May 2021 (unpublished data from our group).

| Lake or Station | Distance (km) | SPM (mg/L) | POC (μmol/L) | DOC (μmol/L) | POC (%) | $\delta^{13}$C (‰) | $\delta^{15}$N (‰) | POC/PN (mol/mol) |
|---|---|---|---|---|---|---|---|---|
| Dongting | 1361 | 21 | 126 | 171 | 7.2 | -26.1 | 5.2 | 6.6 |
| Poyang | 840 | 45 | 169 | 141 | 4.5 | -25.0 | 4.7 | 7.9 |
| Xuliujing | 114 | 40 | 80 | 152 | 2.4 | -23.8 | 6.1 | 6.0 |

As suggested by Reviewer #3, the influence of lakes is very important, and we have added these discussions into the revised manuscript.

References:

Bao, H., Wu, Y., and Zhang, J.: Spatial and temporal variation of dissolved organic matter in the Changjiang: fluvial transport and flux estimation, J. Geophys. Res.: Biogeo., 120, 1870-1886, https://doi.org/10.1002/2015JG002948, 2015.

Bao, H., Wu, Y., Zhang, J., Deng, B., and He, Q.: Composition and flux of suspended organic matter in the middle and lower reaches of the Changjiang (Yangtze River)—impact of the Three Gorges Dam and the role of tributaries and channel erosion, Hydrol. Process., 28, 1137-1147, https://doi.org/10.1002/hyp.9651, 2014.

Liu, X., Li, Y.-L., Liu, B.-G., Qian, K.-M., Chen, Y.-W., and Gao, J.-F.: Cyanobacteria in the complex river-connected Poyang Lake: horizontal distribution and transport, Hydrobiologia, 768, 95-110, https://doi.org/10.1007/s10750-015-2536-2, 2016.

Pan, B.-Z., Wang, H.-J., Liang, X.-M., and Wang, H.-Z.: Factors influencing chlorophyll *a* concentration in the Yangtze-connected lakes, Fresen. Environ. Bull., 18, 1894-1900, 2009.

Wu, Z., Lai, X., Zhang, L., Cai, Y., and Chen, Y.: Phytoplankton chlorophyll *a* in Lake Poyang and its tributaries during dry, mid-dry and wet seasons: a 4-year study, Knowl. Manag. Aquat. Ec., 06, https://doi.org/10.1051/kmae/2013088, 2014.

Second, summer features high discharge and high sediment load, which does not favor autochthonous production (authors also stated this point, e.g., in line 287-288). In fact, a lot of studies have shown high phytoplankton activities in the Yangtze in winter or spring when flow and suspended sediment content are low.

**Response:** Thank you for this comment. Reviewers #1 and #2 also raised similar comments. In the literature, it is generally regarded that higher river discharge in summer should also be accompanied by higher SPM concentrations, largely due to the enhanced soil erosion caused by deforestation (Dai et al., 2016). However, due to recent intensive dam trapping and decreased deforestation in the river basin, the difference in SPM concentration between flood and dry seasons has become much smaller or disappeared (please see Figures D11 and D12, the letter D means the reference of Dai et al. [2016], similarly hereafter). Based on our own data (Figure G4a, and those in Gao et al. [2012]), no clear evidence was found for higher SPM concentrations in flood seasons compared to dry seasons.

[Figure]

Figure D11. Daily SSC (suspended sediment concentration) and water discharge in different flood years: A) 1964; (B) 1973; (C) 1983; (D) 1998; (E) 2010.

[Figure]

Figure D12. Daily SSC and water discharge during different drought years: (A) 1963; (B) 1978; (C) 2011.

[Figure]

Figure G4a. Variations of the monthly average values (±standard deviation) of SPM abundance measured at station #4 during the sampling period September 2009 to August 2010.

References:

Dai, Z., Fagherazzi, S., Mei, X., and Gao, J.: Decline in suspended sediment concentration delivered by the Changjiang (Yangtze) River into the East China Sea between 1956 and 2013, Geomorphology, 268, 123-132, https://doi.org/10.1016/j.geomorph.2016.06.009, 2016.

Gao, L., Li, D., and Zhang, Y.: Nutrients and particulate organic matter discharged by the Changjiang (Yangtze River): Seasonal variations and temporal trends, J. Geophys. Res.: Biogeo., 117, G04001, https://doi.org/10.1029/2012JG001952, 2012.

Third, Xuliujin is the last station before Yangtze enters the coastal ocean. Hence, the tidal influence is significant. Is it possible that autochthonous production in coastal ocean cause the higher $^{13}$C values in summer? How does estuarine process affect organic carbon biogeochemistry of the river?

**Response:** Thank for the comment. As stated in the original manuscript, the salinity values of all our 35 samples never exceeded 0.2, typical values of river waters. In fact, during the flood season in summer, the greatly enhanced Changjiang River discharge would push the freshwater to a much longer distance and dispense river water to a much larger overlying area. According to our own experience, even in downstream areas with a distance of 50 km or more from Xuliujing station, the salinity there was still generally lower than 0.2 regardless of surface water or bottom water in summer. Thus, at the Xuliujing station, the estuarine and tidal process might influence the water levers there, but no saltwater intrusion or influence on salinity and chemical composition occurs there during flood seasons.

(2) The authors conclude on a significant increase of POC, $^{13}$C-POC and $^{15}$N-PN over the past decades based on the literature and their own data, which was attributed to increase in the proportion of autochthonous organic components owing to intensified human activities and global warming in the river basin. However, these data are from three different stations (Datong, Nantong and Xuliujing), and the distance between Datong and Xuliujing stations could be as high as > 500 km. In particular, the Xuliujing station is also likely affected by autochthonous production in the estuary. It seems that the increasing trend of POC and isotopes is likely caused by geographic rather than temporal variations. Also, there is a large gap linking the observed variations with human activities and global warming. I think the above problems are critical to the validness of the conclusions.

**Response:** Thank you for pointing this out. Although the two stations between Datong (for discharges) and Xuliujing (for sampling) are far apart, the total discharges from the several small tributaries between Datong and Xuliujing only accounted for 1.2% of the Datong's annual discharge (Mei et al., 2019). Therefore, the discharge measured at Datong has always been used to represent the ultimate discharge from the Changjiang River.

In terms of the chemical properties, the results from Liu et al. (2003) suggested that the concentrations of all the five nutrients ($NO_3^-$, $SiO_3^{2-}$, $PO_4^{3-}$, $NH_4^+$, and $NO_2^-$) were relatively constant over the whole lower reach of the Changjiang River (please see Figure L2 in Liu et al. [2003]).

[Figure]

Figure L2. Concentrations of nutrients in the main stream of the Changjiang, which are plotted against the distance from the river mouth.

Bao et al. (2015) carried out two samplings (in October 2009 and July-August 2010) in the middle and lower Changjiang mainstream, and they showed that DOC

concentrations between Xuliujing and the nearest upward stations (with distances even longer than that between Datong and Xuliujing) were generally 0.1 mg-C/L (equal to 8.3 μM). Wang et al. (2019) also showed that the DOC and CDOM (quantified by $a_{254}$) over the distance from Xuliujing to stations about 500 km upward were also relatively stable, especially considering the variation ranges after data from the middle and lower reaches had all been included. Similarly Wu et al. (2018) carried out two sampling cruises in October 2009 and August 2010, and they observed that POC (%), $\delta^{13}C$, and conventional ages were rather stable in the SPM samples over the whole lower reach (see their Table W2). Yu et al. (2011) conducted samplings in April-May 2011 and October-November 2006 in the middle and lower reaches of the Changjiang River, and they also found that POC (%) ($0.92 \pm 0.06$ in 2003, and $1.73 \pm 0.23$ in 2006) and $\delta^{13}C$ (‰) values ($-24.90‰ \pm 0.14‰$ in 2003, and $-24.90‰ \pm 0.06‰$ in 2006) were rather similar over the whole lower reach (see their Table Y2). Data from our group also suggested that the chemical properties between Xuliujing and Wuhu (close to Datong) in May 2021 were quite similar (Table R2 below).

Table W2. TSM concentrations (TSM samples only), organic carbon contents (POC%), and bulk stable carbon isotope ($\delta^{13}C$, ‰) and radiocarbon compositions (conventional $^{14}C$ age, years BP) of organic carbon in suspended particulate matter and sediments in the lower reaches of the Changjiang river.

| Sampling period | Station | Water depth (m) | TSM (mg/L) | POC% | $\delta^{13}C$ | Conventional age (BP) |
|---|---|---|---|---|---|---|
| Oct 2009 | Wuhu | 0 | 69.0 | 1.0 | -25.7 | 2190 |
| | | 4.5 | 57.3 | 0.9 | -25.7 | 2200 |
| | | 8 | 61.0 | 0.9 | -25.5 | 1900 |
| | | 13.5 | 55.5 | 0.8 | -25.9 | 2360 |
| | Xuliujing | 0 | 27.2 | 0.8 | -26.0 | 2090 |
| | | 4.5 | 33.7 | 0.8 | -25.4 | 2350 |
| | | 9 | 36.0 | 0.8 | -25.7 | 2250 |

| | | 14.5 | 40.7 | 0.8 | -25.6 | 2240 |
|---|---|---|---|---|---|---|
| Aug 2010 | Jiujiang | 0 | 157.3 | 1.0 | -25.6 | 1570 |
| | | 18 | 173.7 | 1.2 | -25.1 | |
| | Jiangyin | 0 | 136.5 | 1.2 | -25.1 | 2960 |

Table R2. Comparison of chemical properties between Xuliujing and Wuhu stations in the Changjiang River mainstream, collected in May 2021 (unpublished data from our group).

| Station | Distance (km) | SPM (mg/L) | POC (μmol/L) | DOC (μmol/L) | POC (%) | $\delta^{13}C$ (‰) | $\delta^{15}N$ (‰) | POC/PN (mol/mol) |
|---|---|---|---|---|---|---|---|---|
| Wuhu | 492 | 33.7 | 73 | 155 | 2.6 | -23.9 | 5.7 | 7.7 |
| Xuliujing | 114 | 40.1 | 80 | 152 | 2.4 | -23.8 | 6.1 | 7.0 |

In the revised manuscript, we have added some sentences into the revised manuscript, explaining that the samples collected between Datong and Xuliujing were generally similar in chemical properties or no systematic differences/changes could be found between samples collected at these two stations. Our further explanation can also be found in the responses below to your specific comment on Section 4.3.

References:

Bao, H., Wu, Y., and Zhang, J.: Spatial and temporal variation of dissolved organic matter in the Changjiang: fluvial transport and flux estimation, J. Geophys. Res.: Biogeo., 120, 1870-1886, https://doi.org/10.1002/2015JG002948, 2015.

Liu, S. M., Zhang, J., Chen, H. T., Wu, Y., Xiong, H., and Zhang, Z. F.: Nutrients in the Changjiang and its tributaries, Biogeochemistry, 62, 1-18, https://doi.org/10.1023/A:1021162214304, 2003.

Mei, X., Zhang, M., Dai, Z., Wei, W., and Li, W.: Large addition of freshwater to the tidal reaches of the Yangtze (Changjiang) River, Estuar. Coast., 42, 629-640, https://doi.org/10.1007/s12237-019-00518-0, 2019.

Wang, X., Wu, Y., Bao, H., Gan, S., and Zhang, J.: Sources, transport, and transformation of dissolved organic matter in a large river system: Illustrated by the Changjiang River, China, J. Geophys. Res.: Biogeo., 124, 3881-3901, https://doi.org/10.1029/2018JG004986, 2019.

Wu, Y., Eglinton, T. I., Zhang, J., and Montlucon, D. B.: Spatiotemporal variation of the quality, origin, and age of particulate organic matter transported by the Yangtze River (Changjiang), J. Geophys. Res.: Biogeo., 123, 2908-2921, https://doi.org/10.1029/2017JG004285, 2018.

Yu, H., Wu, Y., Zhang, J., Deng, B., and Zhu, Z.: Impact of extreme drought and the Three Gorges Dam on transport of particulate terrestrial organic carbon in the Changjiang (Yangtze) River, J. Geophys. Res.: Earth, 116, F04029, https://doi.org/10.1029/2011JF002012, 2011.

Specific comments
Abbreviations (not full name) should be shown in brackets in the text for the first time.

**Response:** Corrected. Thank you for all the detailed comments.

Line 1-2: the words "variations" and "dynamics" in title are replicate.
Response: Thank you for pointing this out. The title has been modified.

Line 8-10: the abstract did not show research background or scientific questions. Research significance is not shown in the Abstract either.

**Response:** The Abstract has been revised according to your comment. Thank you for this constructive comment.

Line 19-21: the increasing trend need to be reconsidered based on the same station; contribution of autochthonous component to DOC and POC should consider the influence of estuary phytoplankton dynamics and tidal activities.

**Response:** Regarding the potential interference caused by different stations, please see our detailed explanations to your comment on Section 4.3 below. As explained earlier, the chemical properties in waters collected at Xuliujing are seldom influenced by the tidal activities and estuarine phytoplankton dynamics, especially in summer when the Changjiang River discharges were elevated.

Line 39-47: authors stated that biogeochemical cycles of carbon in aquatic environments had long been of great interest in the literature, but did not state the existing work on carbon variations under the influences of climate change and anthropogenic activities. And the knowledge gap deducted from existing work is not clear either in this paper.

**Response:** Thank you for pointing this out. Differentiating the respective influences of climate change and anthropogenic activities is always challenging (Liu et al., 2020;

Lv et al., 2019; Wu at al., 2021). We agree with Reviewer #3 here that a knowledge gap still exists. However, the most important finding of this study was that the significant trends on the decadal time scale are evident and had really occurred over the Changjiang River basin. To the best of our knowledge, the new decadal trends have never been reported for the Changjiang River basin in previous studies. These findings deepened and enriched our knowledge on the ecosystem evolution in the recent decades under the combined effect of global warming and human activities. Unfortunately, based on our current dadaset, we are unable to differentiate the influences of the two factors. Further studies are sorely needed.

References:

Liu, D., Bai, Y., He, X., Chen, C.-T. A., Huang, T.-H., Pan, D., Chen, X., Wang, D., and Zhang, L.: Changes in riverine organic carbon input to the ocean from mainland China over the past 60 years, Environ. Int., 134, 105258, https://doi.org/10.1016/j.envint.2019.105258, 2020.

Lv, S., Yu, Q., Wang, F., Wang, Y., Yan, W., and Li, Y.: A synthetic model to quantify dissolved organic carbon transport in the Changjiang River system: Model structure and spatiotemporal patterns, J. Adv. Model. Earth Sy., 11, 3024-3041, https://doi.org/10.1029/2019MS001648, 2019.

Wu, N., Liu, S.-M., Zhang, G.-L., and Zhang, H.-M.: Anthropogenic impacts on nutrient variability in the lower Yellow River, Sci. Total Environ., 755, 142488, https://doi.org/10.1016/j.scitotenv.2020.142488, 2021.

Line 56-59: these sentences are research methods.
**Response:** We agree. This sentence has been moved to the Methods section.

Section 4.1: the potential factors influencing organic matter quantity and quality were only discussed using their own data when referring to flushing and dilution effects (e.g., lines 263-265, 267-268), but all the others were repeating the conclusions that have been reported by literatures without discussing the major findings of the present study. Therefore, it is hard to tell whether human activities, global warming and autochthonous production did show their effects in the present study. I suggest that the authors concisely summarize their major findings that directly answer their main research questions and focus on explaining and evaluating what they found.

**Response:** Thank you for this suggestion. In this section, our main purpose was to list the possible reasons that may have their influence on the seasonal and decadal trends of organic carbon pools. Some factors, such as global warming, cannot be discussed

based on our own data. We totally agree with you here and we have revised our discussion focusing on what our data can support and on reducing the overall length of text (more concise).

Section 4.2: I agree that the seasonal variations of POC and its isotopes may be related to the ratios of autochthonous to allochthonous components, but I am reserved on that the autochthonous signal of POC at Xuliujing station is related to the upstream reservoir constructions (e.g., the Three Gorges Dam) which is like >1500 km far away. What about the influences of autochthonous production in upstream lakes (e.g., Dongting and Poyang Lakes)? And is it possible that the POC quantity and quality are influenced by the phytoplankton dynamics in the estuary or coastal ocean where autochthonous production is strong in summer?

**Response:** There is no dispute that the constructions of more than 50,000 dams over the recent decades, including the Three Gorges Dam, are the main reason that had led to the sharply decreasing SPM concentrations and transport fluxes (Yang et al., 2011). As pointed out by Dai et al. (2016), the river waters were much "cleaner" than before, even in the lower reach of the Changjiang River. Similar phenomenon not only occurred in the Changjiang River but also in the Yellow River in China (Wang et al., 2017). Thus, the river water ecosystems in the lower reach should also have been changed in response to the decreased SPM, largely due to the daming effects.

Regarding the effects of lakes, please see our response to your general comment.

Regarding the possible effect from the estuarine processes, please see our response to your general comment.

References:

Dai, Z., Fagherazzi, S., Mei, X., and Gao, J.: Decline in suspended sediment concentration delivered by the Changjiang (Yangtze) River into the East China Sea between 1956 and 2013, Geomorphology, 268, 123-132, https://doi.org/10.1016/j.geomorph.2016.06.009, 2016.

Wang, H., Wu, X., Bi, N., Li, S., Yuan, P., Wang, A., Syvitski, J. P. M., Saito, Y., Yang, Z., and Liu, S.: Impacts of the dam-orientated water-sediment regulation scheme on the lower reaches and delta of the Yellow River (Huanghe): A review, Global Planet. Change, 157, 93-113, https://doi.org/10.1016/j.gloplacha.2017.08.005, 2017.

Yang, S. L., Milliman, J. D., Li, P., and Xu, K.: 50,000 dams later: Erosion of the Yangtze River and its delta, Global Planet. Change, 75, 14-20, https://doi.org/10.1016/j.gloplacha.2010.09.006, 2011.

Line 309-311: is there any data supporting the source of POC from deep soils? Why are waters from deep flow paths high in DOC concentration? Can belowground water influence DOC concentration?

**Response:** This sentence has been deleted from the revised manuscript since we do not have relevant data to support this. Thank you for pointing this out.

Section 4.3: authors tried to show the decadal trend of SPM, DOC, POC and their isotope signals using reported data and literature data. This is a great idea, however, these data are from three different stations (Datong, Nantong and Xuliujing), of which the Xuliujing station is likely severely influenced by estuary phytoplankton dynamics and tidal activities. More importantly, the increasing trend of POC, $^{13}$C-POC and $^{15}$N-PN in Figure 11 is very likely caused by different stations (higher values in Xuliujing station) instead of time. I'm afraid that the decadal trends need to reconsideration.

**Response:** As we have explained earlier, the chemical properties between Datong and Xuliujing, even over the entire lower reach of the Changjiang River, were generally similar. No systematic changes in chemical property parameters between Datong and Xuliujing could be found, as reported in previous studies. In fact, if the data from Datong were removed from Figure 11 (the two stations of Xuliujing and Nantong are very close), the increasing trends would be still significant and seemed to be even stronger than before (Figure R1), and the decadal trends would not be changed by this modification, again suggesting that the significant decadal trends were not caused by the spatial factor over the distance between Datong and Xuliujing.

[Figure]

Figure R1. Variations of POC (%), DOC/POC ratios, δ¹³C, δ¹⁵N, and POC/PN ratios in the lower Changjiang River between 1993 and 2019. **In this figure, the data obtained from the Datong station have been removed and only data from Xuliujing and Nantong remain.**

Figure 1: add dam positions.

**Response:** Done. Thank you for this comment.

Figures 9 and 10: when investigating temporal variations, using data from the same station should be more compellent.

**Response:** We agree with you here. In order to collect sufficient data and to make our study more representative, using additional data from other stations is a better choice (note that all these stations are restricted in the lower reaches from Datong to Xuliujing). It should also be emphasized that even if we removed the data at the Datong station, our conclusions would still hold and not be altered. Please see our response to previous comments.

Figure 11: add legend of points including station name

**Response:** Done. Thank you for this comment.

Again, we appreciate the Reviewer for the constructive and insightful comments and time spent on our manuscript. The comments have greatly improved our manuscript. We hope that our revised manuscript now meets the standard set by *Biogeosciences*.